# LOCATE-THEN-EDIT FOR MULTI-HOP FACTUAL RECALL UNDER KNOWLEDGE EDITING

## ABSTRACT

The locate-then-edit paradigm has shown significant promise for knowledge editing (KE) in Large Language Models (LLMs). While previous methods perform well on single-hop fact recall tasks, they consistently struggle with multi-hop factual recall tasks involving newly edited knowledge. In this paper, leveraging tools in mechanistic interpretability, we first identify that in multi-hop tasks, LLMs tend to retrieve implicit subject knowledge from deeper MLP layers, unlike single-hop tasks, which rely on earlier layers. This distinction explains the poor performance of current methods in multi-hop queries, as they primarily focus on editing shallow layers, leaving deeper layers unchanged. To address this, we propose IFMET, a novel locate-then-edit KE approach designed to edit both shallow and deep MLP layers. IFMET employs multi-hop editing prompts and supplementary sets to locate and modify knowledge across different reasoning stages. Experimental results demonstrate that IFMET significantly improves performance on multi-hop factual recall tasks, effectively overcoming the limitations of previous locate-then-edit methods.

## 1 INTRODUCTION

Large Language Models (LLMs) like ChatGPT (Achiam et al., 2024) and LLaMA-2 (Touvron et al., 2023) have emerged as powerful knowledge bases, demonstrating remarkable abilities in both factual knowledge representation and reasoning over complex queries (Etezadi & Shamsfard, 2022). However, as the need for updating and correcting knowledge within these models grows, research on knowledge editing (KE) has gained significant attention, focusing on cost-effective ways to modify specific information in LLMs (Mazzia et al., 2023). KE methods can be broadly classified into two categories based on whether they alter the original model weights: weight-preserving (Zhong et al., 2023) and weight-modifying approaches (Meng et al., 2022a;b). Weight-preserving methods aim to modify the model's outputs by integrating external memory or leveraging strategies such as in-context learning without altering the underlying weights (Cheng et al., 2024b;a). In contrast, weight-modifying methods directly change the model's internal weights to update the stored knowledge. Weight-modifying methods can be further categorized into learning-based and optimization-based methods. Learning-based methods update weights using gradients but face challenges such as overfitting and poor generalization. Optimization-based methods, such as ROME (Meng et al., 2022a) and MEMIT (Meng et al., 2022b), have introduced the "locate-then-edit" paradigm, which first identifies the knowledge storage layers and then adjusts their weights through optimization techniques to achieve the desired knowledge modification.

Compared to weight-preserving methods and learning-based weight-modifying approaches, the locate-then-edit paradigm offers precise editing of the model's internal knowledge with low computational costs (Zhang et al., 2024). However, despite the success of locate-then-edit methods in single-hop fact recall tasks (Li et al., 2024c), they share a common limitation Zhong et al. (2023): **The post-edited model struggles with multi-hop factual recall tasks involving the newly edited knowledge** (see Table 3 for details). For example, after changing the knowledge "The capital of Spain" from "Madrid" to "Hartford", the model correctly answers $Q_1$ = "What is the capital city of Spain?". However, when posed with the multi-hop question $Q_2$ = "What is the capital city of the country where Pablo Picasso holds citizenship?", it still responds with "Madrid" (Figure 1 (b)). This discrepancy raises a natural question: *Has the locate-then-edit approach reached its limits for multi-hop factual recall tasks, or does it still hold unexplored potential?*

Figure 1: **(a)** The existing locate-then-edit KE method updates **new fact** to the shallow layers of the model using a single-hop edit template. **(b)** For multi-hop fact recall tasks, especially when the edited fact is in the second or subsequent hops, the hops typically access the deeper layers which outputs the **unmodified knowledge**. **(c)** Our method introduces a **prefix hop** for each single-hop edit, creating a two-hop edit template. We utilize this new template to perform a furtherance edit, targeting the deeper layers for more effective knowledge updating.

To answer this question, we first explored the mechanisms of the pre-edited model when handling single-hop and multi-hop factual recall tasks to gain insights. Using the example mentioned, we attempt to illustrate how the model reasons with the implicit subject "Spain" in $Q_2$, compared to the explicit mention in $Q_1$. We first use LogitLens (nostalgebraist, 2020; Dar et al., 2023) to interpret the information encoded in each layer's hidden states by projecting them into the output vocabulary space. We find that at the last token position, the information of the implicit subject accumulates before the final answer, which is significantly different from the single-hop scenario. We then conduct causal intervention experiments (Li et al., 2024d) to further confirm the influence of the implicit subject on the final answer.

Based on this, we further explore the mechanism of how the implicit subject influences the prediction of the final answer. By using causal intervention, our results indicate that in the multi-hop scenario, the implicit subject guides the emergence of the final answer by retrieving relevant knowledge from the **later MLP layers**. This contrasts sharply with the single-hop cases (Meng et al., 2022a; 2023), where the subject information is used to retrieve information from **earlier MLP layers**. Based on this difference, we provide an explanation for the unsatisfactory performance of the existing locate-then-edit methods for multi-hop tasks: Previous methods leveraging single-hop prompts for editing are insufficient as they only update the relevant knowledge in the shallow MLP layers but fail to propagate the changes to deeper layers. The model retains some of the old single-hop knowledge that only activated by additional implicit multi-hop fact recall mechanisms.

Based on these observations, we developed an advanced locate-then-edit KE method specifically designed to modify knowledge in both shallow and deep MLP layers, which we named **I**nterpretability-Guided **F**urtherance **M**odel **E**diting in a **T**ransformer (**IFMET**). IFMET introduces a supplementary set for edit instances and generates multi-hop editing prompts, surpassing the limitations of single-hop prompts used in previous locate-then-edit approaches. This supplementary set helps us locate pre-existing knowledge that appears in later hops by leveraging the differences between the reasoning mechanism for single-hop and multi-hop queries. By leveraging each edit instance and its corresponding multi-hop editing prompt, IFMET locates and edits the knowledge stored in both earlier and later MLP layers, effectively addressing cases where the knowledge to be edited appears either in the first or subsequent hops during reasoning, as illustrated in Figure 1. Our contributions can be summarized as follows:

- We first identified key differences in the mechanisms the model uses for reasoning in single-hop versus multi-hop fact recall tasks. In multi-hop scenarios, unlike single-hop cases, the model prioritizes inferring the implicit subject at the last token position, which guides the generation of the final answer.
- Next, we pinpointed the components of the implicit subject that influenced the final answer within the later MLP layers. We demonstrated that the absence of edited knowledge of these components significantly impacted the model's performance.

- We propose IFMET, an advanced locate-then-edit KE method specifically designed to modify knowledge in both shallow and deep MLP layers using single and multi-hop edit prompts. Experimental results confirm the effectiveness of our method, showing that it successfully overcomes the limitations of previous locate-then-edit approaches in handling multi-hop factual recall tasks.

Due to the space limit, we refer readers to Appendix A for previous work.

## 2 PRELIMINARIES

**Notations.** We define the set of knowledge as $\mathcal{K} = \{(s, r, o)\} \subseteq \mathcal{E} \times \mathcal{R} \times \mathcal{E}$, where $\mathcal{E}$ and $\mathcal{R}$ denote the set of entities and relations respectively. Each tuple $(s, r, o) \in \mathcal{K}$ represents that the corresponding entity of subject entity $s$ under relation $r$ is object entity $o$. An editing instance can be described in the form of a triplet: $e = (s, r, o \rightarrow o^*)$, where $o^*$ denotes the new edited object in place of the original object $o$ related to $s$ through $r$.

*Multi-hop factual recall $Q$* requires multi-step reasoning to reach the final answer. Its reasoning process is composed of a chain of knowledge $C = (s_1, r_1, o_1) \oplus \cdots \oplus (s_n, r_n, o_n)$, where $s_1$ is the start subject that is explicitly given in the question, $o_n$ is the final answer, and $\oplus$ used for chain adjacent reasoning steps which means the subject $s_{i+1}$ is identical to the object $o_i$ of preceding reasoning step. In order to better explore how the language model recalls multi-hop questions, we categorize the reasoning step into two types: *explicit recall step* $(s_1, r_1, o_1)$ and *implicit recall steps* $\{(s_2, r_2, o_2), \ldots, (s_n, r_n, o_n)\}$. The inference information required by the former subject $s_1$ explicitly appears in the prompt, while the subjects of the latter $s_2...s_n$ need to be inferred to obtain, which are called implicit subjects.

### 2.1 FACTUAL RECALL TASKS

**Format of Factual Recall Tasks.** Factual recall tasks refer to verifying whether the model $\mathcal{M}$ can correctly provide the final answer to a single-hop question or a multi-hop factual recall $Q$. Based on the two forms of declarative sentences and interrogative sentences, there are two different formats of factual recall tasks: Cloze-Format $Q_{cloze}$ and QA-Format $Q_{qa}$. For instance, given two-hop questions with the knowledge chain like *(Paradiso, author, Dante Alighieri) $\oplus$ (Dante Alighieri, country of citizenship, Italy)*, $Q_{cloze}$ can be "The author of Paradiso is a citizen of", while $Q_{qa}$ is "What country does the author of Paradiso hold citizenship in?". If the model's final answer is the same as the answer to the question, the recall is considered successful, which can be represented as $\mathcal{M}(Q_{cloze}) = o_n$ or $\mathcal{M}(Q_{qa}) = o_n$.

**Multi-hop Factual Recall under Knowledge Editing.** This task assesses whether the post-edited model can effectively leverage the updated knowledge for reasoning in multi-hop fact recall tasks. Given an edit $e = (s, r, o \rightarrow o^*)$, the edit prompt $T_r(s)$ and a chain of facts $C_e$ which includes $(s, r, o)$ as one of its components. After the post-edited model must leverage the new factual knowledge $(s, r, o^*)$ to answer the multi-hop query. For example, given edit *(Paradiso, author, Dante Alighieri $\rightarrow$ Mark Twain)*, the model's response of "The author of Paradiso is a citizen of" should change from the original answer *Italy* to the new answer *USA*.

### 2.2 MECHANISTIC INTERPRETATION TOOLS

**LogitLens.** LogitLens (nostalgebraist, 2020) is a framework for interpreting the hidden states (activations) of language models such as GPT (Brown et al., 2020) by examining the logits (the raw prediction scores before they are transformed into probabilities) and corresponding probabilities. Specifically, for the hidden state $h_l^i$ at the $l$-th layer and position $i$, the logits $s_l^i$ and probabilities $p_l^i$ over the output vocabulary set $V$ are defined as follows:

$$\begin{cases} s_l^i = W_U h_l^i \in \mathbb{R}^{|V|}, \\ p_l^i = \text{softmax}\left(s_l^i\right) \end{cases}$$

where $W_U$ denotes the unembedding matrix, which is the same matrix used in the final layer of the model for prediction. LogitLens aids in the decomposition of model predictions, elucidating the contributions from various input components such as MLPs and attention heads. This decomposition can be explored by modifying $h_l^i$ to the output from MLP $m_l^i$ or attention heads $a_l^i$, where

$h_l^i = h_i^{l-1} + m_l^i + a_l^i.$ [1] LogitLens posits that probabilities and logits provide insights into how the model prioritizes different potential tokens, as indicated by the proportion of related information. Specifically, we define $\text{Info}(h_l^i, j)$ as the information related to token $j \in V$ contained in $h_l^i$, positively correlated with $s_l^i[j]$ and $p_l^i[j]$. To account for the probability variations across different layers, we define $\text{Info}(h_l^i, j)$ as the layer-wise min-max normalized probability (Li et al., 2024d), where $L$ is the total number of layers:

$$\begin{cases} p_{max}^i[j] = \max_{\{l=1,\ldots,L\}} p_l^i[j], \\ p_{min}^i[j] = \min_{\{l=1,\ldots,L\}} p_l^i[j], \\ \text{Info}(h_l^i, j) = \frac{p_l^i[j] - p_{min}^i[j]}{p_{max}^i[j] - p_{min}^i[j]} \end{cases}$$

**Causal Intervention on Hidden States.** Causal intervention on hidden states Li et al. (2024d;a) involves deliberately altering specific hidden states in a model to observe the resulting changes in various metrics, thereby helping to establish cause-and-effect relationships. This process includes three pivotal components: the intervention operation $\mathcal{I}$ to be conducted, the target hidden state $\mathcal{H}$ selected for intervention, and the effect metric *IE* which measures the change caused by the intervention $\mathcal{I}$. In this paper, the possible hidden states $\mathcal{H}$ for intervention include the layer hidden states $h$, the output hidden states from MLPs $m$, and the output hidden states from attention heads $a$. We use the change in probability $p_l^i[j]$ from LogitLens as the effect metric *IE*, which quantifies the change in the probability of predicting the target token $j$ at layer $l$ for a specific position $i$. This metric enables us to determine whether specific components or tokens, have a causal influence on the model's predictions.

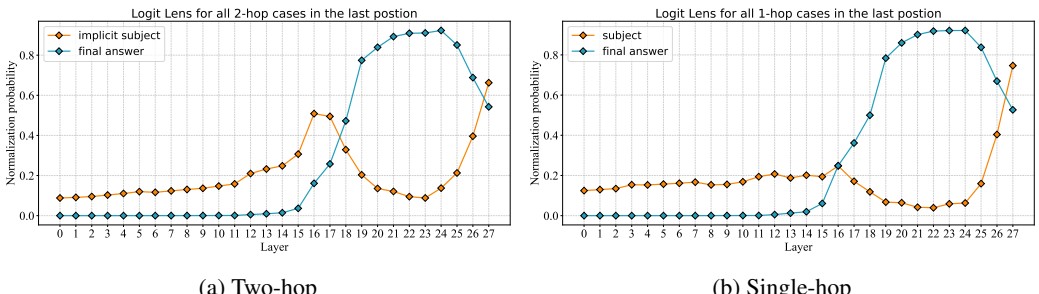

(a) Two-hop                          (b) Single-hop

Figure 2: **LogitLens results of the last token position at different layers.** (a) Yellow line represents the information containing implicit subject $s_2$, i.e., $\text{Info}(h_l, s_2)$. Blue line represents the information for the final answer, i.e., $\text{Info}(h_l, o_2)$. (b) Yellow line represents the information of subject $s$. i.e., $\text{Info}(h_l, s)$ and Blue line represents the information of the answer $o$, i.e., $\text{Info}(h_l, o)$. Larger versions of the sub-figures are available in the Appendix 8b.

## 3 MECHANISMS OF KNOWLEDGE STORAGE AND REASONING

In this section, we will explore the reasoning mechanisms of the pre-edited model for both single-hop and multi-hop factual recall tasks. By comparing the knowledge utilization processes, we identify the reasons behind the suboptimal performance in multi-hop tasks and explain why the post-edited model tends to output the original answer instead of the new edited one. Specifically, we focus on two-hop tasks to better illustrate these distinctions. Experiments are conducted using a subset of single and two-hop data from MQuAKE-CF (Zhong et al., 2023) with the GPT-J (6B) model (Wang & Komatsuzaki, 2021). More detailed information about the data and experimental setup is provided in Appendix B.1.1.

### 3.1 HOW THE PRE-EDITED MODEL REASONS FACT RECALL TASKS

For a multi-hop fact recall task, the knowledge chain is represented as $C = (s_1, r_1, o_1) \oplus \cdots \oplus (s_n, r_n, o_n)$. The model may employ multiple strategies to answer such tasks, including the for-

---

[1]We employ GPT variants such as GPT-J Wang & Komatsuzaki (2021) that position attention in parallel to the MLP, which mathematically equates to models that calculate MLP sequentially after the attention module, as discussed in Brown et al. (2020).

mation of a single super-relation (Ju et al., 2024) $(s_1, r_{mul}, o_n)$, where $r_{mul} = r_1 \rightarrow \cdots \rightarrow r_n$, or by segmenting the task into one explicit recall step followed by several implicit recall steps to answer step-by-step. Previous research (Hou et al., 2023) suggests that models typically engage in reasoning by considering each single-hop recall individually.

Based on this understanding, we hypothesize that the model will prioritize deducing the implicit subjects $\{s_2, \ldots, s_n\}$ and subsequently recall the final answer $o_n$ based on the last implicit subject $s_n$. The subsequent sections aim to verify this hypothesis by examining the model's behavior in structured multi-hop fact recall tasks.

**Interpretation via Hidden Representations.** We use LogitLens to examine the accumulation of information related to the implicit subject $s_2$ and the final answer $o_2$ in the two-hop scenario. The model's predictions for $o_2$, are derived from the last token of the prompt, where crucial information about the resolved implicit subject $s_2$ should be propagated (Biran et al., 2024). Therefore, we focus on the hidden state $h_l$ at the $l$-th layer of the last token position, analyzing $\text{Info}(h_l, s_2)$ and $\text{Info}(h_l, o_2)$ as measures of the information related to $s_2$ and $o_2$ contained in $h_l$. Intuitively, these metrics quantify how much information about $s_2$ and $o_2$ accumulates in the hidden state. The results, depicted in Figure 8a, show that $\text{Info}(h_l, s_2)$ gradually reaches its peak during middle layers [15-17], while $\text{Info}(h_l, o_2)$ increases and peaks during later layers [21-23]. This pattern suggests that, in multi-hop tasks, the implicit subject $s_2$ is processed during the middle layers before reaching the final answer $o_2$.

To explore if single-hop fact recalls $(s, r, o)$ follow the same trend as in multi-hop cases, we conducted a similar experiment using LogitLens. The results, shown in Figure 2b, indicate that $\text{Info}(h_l, s)$ significantly increases after layer 24 and peaks at layer 27, whereas $\text{Info}(h_l, o)$ consistently reaches its peak during layers 21,22,23. This finding implies that there is no significant peak for the subject information before the final answer probability begins to accumulate, suggesting that the accumulation process of the final answer in single-hop cases may not be significantly correlated with the subject information at the last token.

> **Takeaway 1**
>
> In multi-hop scenarios, the implicit subject information consistently accumulates before the final answer at the last token position. However, in single-hop scenarios, since the subject is explicitly given, there is no need for accumulation at the last token position.

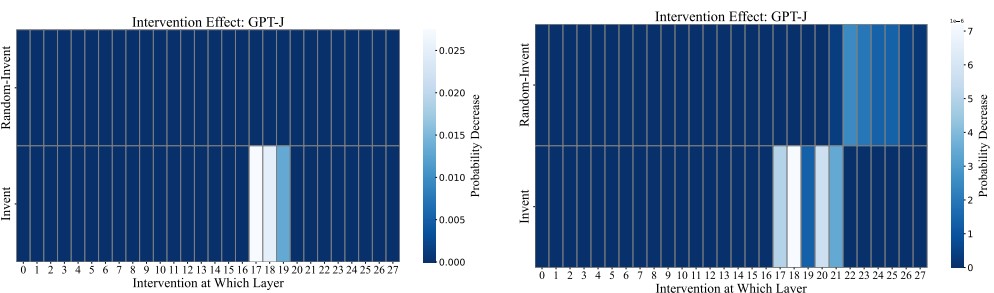

(a) Causal intervention results of layer hidden state in last token position.

(b) Causal Intervention result of MLP hidden state in last token position

Figure 3: **Causal Intervention Result**: A brighter color signifies a stronger intervention effect. In each subfigure, upper row represents experimental group, while upper row is control group. Note that negative effect values ($\leq 0$) are clipped to 0 in both groups for better visualization. (a) is probability change $IE_h$ of intervention $\mathcal{I}_h$, (b) is probability change $IE_m$ of intervention $\mathcal{I}_m$.

**Causal Intervention.** Next, we explore whether the appearance of $s_2$ guides the subsequent information accumulation process of the final answer $o_2$. To this end, we aim to identify which layers facilitate this influence. We propose an intervention experiment where we reduce the information content of $s_2$ at the last token position and observe the changes in the output probability of the final answer in the last prediction layer.

Specifically, we replace the hidden state $h_l$ in layer $\ell$ of the last token with $h_l^*$, and the corresponding logits $s_l\ (= W_U h_l)$ and $s_l^*\ (= W_U h_l^*)$ for $h_l$ and $h_l^*$, respectively. $s_l^*$ is defined as:

$$s_l^*[j] = \begin{cases} \min(s_l[j]), & \text{if } j \in s_2 \\ s_l[j], & \text{otherwise,} \end{cases} \tag{1}$$

where we minimize the logits corresponding to the tokens in $s_2$ without altering the values of other tokens, aiming to diminish the effect of $s_2$. This setup allows us to describe the process through a causal intervention framework, where the intervention $\mathcal{I}_h$ and the effect $IE_h$ are defined as follows:

$$\mathcal{I}_h : h_l^* = h_l + \underset{\Delta h_l}{\arg\min} \| W_U(h_l + \Delta h_l) - s_l^* \|^2, \quad IE_h = p_L[j] - p_L^E[j], \quad j \in o_2, \tag{2}$$

where $L$ is the last layer, $p_L[j]$ denotes the original output probability of $o_2$ in the $L$-th layer, and $p_L^E[j]$ is the probability after the intervention is applied. This approach illustrates how the hidden states and probabilities are expected to change when the logits are modified to $s^*$. For computational efficiency, we opt to approximate $h_l^*$ using a combination of least squares and minimum-norm methods (Lawson & Hanson, 1995) (further details are provided in Appendix B.2).

For comparison, we also randomly select an irrelevant token $j \notin s_2 \cup o_2$ to execute the intervention as the control group. Figure 3a presents the outcomes of our intervention experiments across all layers, where a brighter color signifies a stronger intervention effect. We found a clear positive impact from intervening in layers [17-19] for the experimental group, in contrast to no significant effects observed in the control group across all layers. This suggests that the information of $s_2$ encoded in the intermediate layers plays a crucial role in the probability accumulation process of $o_2$. We also do the same causal intervention experiments for single-hop fact recall (see Appendix B.3 for the results). However, the results indicate that the prediction of $o$ does not significantly rely on the subject information $s$ in the single-hop fact recall.

> **Takeaway 2**
>
> Unlike the mechanism of reasoning the knowledge in single-hop scenarios, in the reasoning process of the second-hop knowledge in two-hop scenarios, the accumulated subject information has causal effects on the final answer, guiding the extraction of related knowledge in the last layer.

**Intermediate Reasoning Results Influence the Knowledge Extraction from MLP.** As previous studies claimed that single-hop tasks retrieve subject information from MLP layers (Meng et al., 2022a;b), we will focus on MLP layers to further investigate the specific mechanisms to answer how the implicit subject $s_2$ influences the prediction of the final answer $o_2$. We conducted a causal intervention experiment similar to the experiments above but focused specifically on the MLP component. Specifically, we aim to replace $m_l$ (the output hidden state of the last token in the $l$-th MLP layer) with $m_l^*$, where we have $s_l = W_U m_l$ and $s_l^* = W_U m_l^*$ with $s_l^*$ is same as in (1). The intervention $\mathcal{I}_m$ shares the same idea as in (2), except that $h_l$ is replaced with $m_l$. However, we redefine the intervention effect $IE_m$, which differs from the previous $IE_h$. In detail, we no longer use the probability at the final layer as the metric; instead, we use the probability calculated from the output of MLP at the modified layer $l$. In total, our causal intervention is formulated as

$$\mathcal{I}_m : m_l^* = m_l + \underset{\Delta m_l}{\arg\min} \| W_U(m_l + \Delta m_l) - s_l^* \|^2, \quad IE_m = p_l[j] - p_l^E[j], \quad j \in o_2.$$

Figure 3b presents the outcomes of our intervention experiments across all layers with the similar control group as in the above.[2] The clear positive impact from intervening in the intermediate layers [17-21] is demonstrated in the experimental group, in contrast to negligible effects observed in the control group across all layers. This suggests that the implicit subject $s_2$ at the last token position was used for retrieving the related information of $o_2$ from **later MLP layers**. Thus, it plays an important role in the probability accumulation process of $o_2$. Note that this is in contrast with previous work (Meng et al., 2022a; 2023), which mentioned that explicit single-hop tasks primarily rely on the subject position token to retrieve information from **earlier MLP layers**.

---

[2]Note that, considering the tiny output probability of MLP, we did not use normalization of probability changes here.

> **Takeaway 3**
>
> During the reasoning process of the second-hop knowledge in two-hop scenarios, information related to the subject is used for retrieving relevant knowledge of the final answer from later MLP layers of the last token position, which is from the earlier MLP layers in single-hop cases.

## 3.2 Why Existing locate-then-edit KE Methods Failed

Based on the findings above, we can provide an explanation for the unsatisfactory performance of the existing locate-then-edit methods. For an editing instance $(s, r, o \rightarrow o^*)$, using only the corresponding explicit single-hop prompt for editing is insufficient as previous methods only update the relevant knowledge in the shallow MLP layers but fail to propagate the changes to deeper layers, which is utilized in multi-hop fact recall tasks.

Table 1: Comparison of QA and Cloze Formats for $D_{Pre}$ and $D_{Post}$

| Edit Batch | QA Format(%) ↑ | | Cloze Format(%) ↑ | |
|---|---|---|---|---|
| | $D_{Pre}$ | $D_{Post}$ | $D_{Pre}$ | $D_{Post}$ |
| GPT-J | 50.62 | 41.72 | 20.31 | 18.63 |
| Edit=1 | **64.29** | **2.93** | **43.37** | 4.60 |
| Edit=100 | 63.27 | 3.35 | 42.86 | **3.35** |

We provide a concrete example for a better understanding. Given an editing instance $(Spain, captical, Madrid \rightarrow Hartford)$, and $Q_{cloze}(s)$ is "The capital city of Spain is". Existing methods modify the weights of shallow MLPs with $Q_{cloze}(s)$ to make it answer *Hartford*. The paradigm may be well-suited for cases where the modified information is queried in a single-hop manner, as these tasks retrieve answers from the early MLP layers. However, it will be ineffective when the modified knowledge is queried in the second or later fact recall steps, where the model relies on deeper MLP layers at the last token position for knowledge retrieval. In this example, the first-hop query "The capital city of Spain is located in the continent of" should be answered correctly because it retrieves the knowledge $(Spain, captical, Hartford)$ in shallow MLPs. However, the second one "The capital city of the country has nationals Pablo Picasso is" is still answered with *Madrid* because the knowledge $(Spain, captical, Madrid)$ stored in later MLPs does not changed.

To verify our above claim, we divide two-hop fact recall tasks into two sets $D_{Pre}$ and $D_{Post}$, depending on the position of the edited knowledge within the two-hop reasoning process. Specifically, for an edited knowledge $(s, r, o, o*)$, we have the following two sets after editing.

$$D_{Pre} = \{(s, r, o^*) \oplus (s_2, r_2, o_2)\}, \quad D_{Post} = \{(s_1, r_1, o_1) \oplus (s, r, o^*)\}.$$

We sampled two subsets with approximately equal size from the MQuAKE-CF dataset, detailed in the Appendix B.1.2. By applying the SOTA locate-then-edit method PMET to layer [3-8], which follows (Li et al., 2024c), we present the percentage of cases where both pre-edited and post-edited models answer successfully in QA format or Cloze format under different edit batches.

Table 1 shows the results of the comparative experiments. We can see that performance on $D_{Pre}$ is significantly better than on $D_{Post}$, which aligns with our expectations. This is because reasoning the first hop knowledge in $D_{Pre}$ is similar to the single-hop process. After updating the knowledge in the earlier MLP layers, the model is likely to effectively use the newly edited knowledge. It can use the new implicit subject in the second hop to produce the final updated answer. When facing cases in $D_{post}$, PMET cannot get the correct final answer because it only modifies the earlier MLP layers, which is not enough for the model to correctly reason the second hop knowledge as it should be retrieved from later MLP layers.

## 4 IFMET: An Advanced Locate-then-Edit Method

Motivated by our findings on the distinctions between single-hop and multi-hop factual recall process, we introduce the Interpretability-Guided Furtherance Model Editing in a Transformer (IFMET). This method addresses the limitations identified in existing locate-then-edit approaches by modifying knowledge across both earlier and later MLP layers, enhancing the model's ability to handle multi-hop reasoning. The IFMET method comprises two main steps: first, constructing a supplementary set of original edits to enrich the edit context, and second, perform-

ing editing based on multi-hop prompts derived from the original edit case and its supplementary set. This furtherance step approach ensures a thorough integration of new knowledge, significantly improving the model's accuracy and robustness in multi-hop factual recall scenarios.

| $e$ | (Spain, capital, Madrid $\rightarrow$ Hartford) |
|---|---|
| $s$ | Spain |
| $T_r(s)$ | The capital city of Spain is |
| $C$ | (Manuel Almunia, citizenship, Spain) (Spain, capital, Madrid $\rightarrow$ Hartford) |
| $Q_C(s)$ | What is the capital city of the country where Manuel Almunia holds citizenship? |
| $e'$ | (Barcelona, country, Spain) |
| $s'$ | Barcelona |
| $C'$ | (Barcelona, country, Spain) (Spain, capital, Madrid $\rightarrow$ Hartford) |
| $T_C(s')$ | The capital city of the country where Barcelona is located is |

Table 2: An support case for an instance in the MQuAKE-CF dataset and the corresponding additional support cases are shown in the lower part.

**Supplementary Set Construction.** Note that for a given edit $e = (s, r, o \rightarrow o^*)$ (it can be extended to cases involving multiple edited facts), a locate-then-edit algorithm typically aims to identify and modify the knowledge-storing MLPs. Previous efforts have predominantly focused on the earlier MLP layers; however, our findings indicate that such an approach underperforms when the edited knowledge appears in second or subsequent hops during reasoning. Given that each edit traditionally targets single-hop knowledge, our experiments have demonstrated that using such edit prompts alone does not effectively update the later knowledge-storing MLPs. To address this issue, we construct a supplementary set for each edit, designed to facilitate the modification of deeper MLPs that provide knowledge in implicit fact recall steps.

In our supplementary set, we transform each edit into a multi-hop chain. For instance, for an edit $e = (s, r, o \rightarrow o^*)$, we can create a supplementary fact $e_{\text{sup}} = (s', r', o')$ where $o' = s$, forming a two-hop fact recall chain $C = (s', r', o') \oplus (s, r, o)$. This approach enables us to subsequently target and modify the latter MLPs that store the fact $(s, r, o)$, updating the information to $(s, r, o^*)$. An illustrative example of this process is provided in Table 2.

Practically, we utilize WikiData[3] to construct the supplementary dataset. We start by extracting all subjects from the dataset's edits and deduplicating them to form a set of subjects $S_e = \{s_i | i = 1, \dots \}$. We then perform a WikiData SPARQL query[4] to identify a set of triplets for each subject $s_i$: $Sup = \{(s', r', o') | o' = s_i\}$. To ensure the reliability of these facts, we filter out examples that cannot be correctly answered using the few-shot approach proposed by (Zhong et al., 2023). For construction details, please refer to the Appendix C.

**Interpretability-Enhanced Furtherance Model Editing in a Transformer.** Now we introduce the proposed IFMET framework. Each pre-edited knowledge has an additional multi-hop chain, assisted by the supplementary set. Based on the difference between the single and multi-top settings we discussed above, we have to locate and modify weights in both earlier and later layers in MLPs.

Based on the previous key-value memories Geva et al. (2021), our method is based on the hypothesis that factual knowledge is stored within the Feedforward Neural Networks (FFNs) of MLPs. Specifically, for the $l$-th layer FFN of the $i$-th token, its output is given by: $v_l^i = f(W_l^{in} h_{l-1}^i)W_l^{out}$, where $f(\cdot)$ is the activation function, and $h_{l-1}^i$ is the input of the $l$-th MLP layer (for simplicity, the superscript $l$ is omitted in the following discussion). In this context, $f(W^{in}h^i)$ functions as the keys, denoted as $k_i$, the outputs of the subsequent layer represent the corresponding values $v_i$, and $W^{out}$ denotes the weights of the knowledge stored in the FFN that needs modifying. Such a structure is well aligned with the triplet form in a fact $(s, r, o)$, where the keys $k_i$ correspond to entities of interest $s_i$ or some specific fact $(s_i, r_i)$ and values $v_i$ contain information about $o_i$. Thus, we have $W^{out}k = v$ for $(k, v)$, which represents the fact $(s, r, o)$ (Geva et al., 2021). We aim to modify $W^{out}$ such that $W^{out}k = v^*$, where $v^*$ contains the information of the new knowledge.

Motivated by the above, in IFMET, considering a modification, there are two steps for both earlier and latter layers in MLPs: $Search$ and $Calculate$. The $Search$ process identifies the suitable $v^*$ through the edit prompt corresponding to the triplet. Then the $Calculate$ process computes the change in weights $W^{out}$ using $v^*$. These two processes are foundational in existing knowledge editing methodologies. In experiments, we adopt the state-of-the-art locate-then-edit method PMET

---

[3] www.wikidata.org

[4] https://query.wikidata.org/

(Li et al., 2024c). The primary differences between the first and further edits are reflected in the edit prompt and the layers edited. Specifically, for the edit instance $e = (s, r, o \rightarrow o*)$, the first edit utilized a one-hop edit template $T_r(s)$ provided by MQuAKE to edit early layers of the model. For the furtherance edit, a two-hop template $T_C(s')$ composed of a support case $(s', r, s)$ and $(s, r, o^*)$ was used, and this template was applied to edit later layers of the model. Due to space limitations, the flowchart of the algorithm and related implementation details are provided in Algorithm 1 and Appendix C.

## 5 EXPERIMENTS

### 5.1 EXPERIMENTAL SETUP

**Dataset.** We use MQuAKE-3K (Zhong et al., 2023), a challenging dataset designed to evaluate models' ability to perform multi-hop reasoning with newly edited knowledge. Each entry consists of multiple single-hop edits and includes multi-hop reasoning questions.

**Baselines.** As **IFMET** is a locate-then-edit approach, we mainly compare it with previous weight-modifying approaches. Specifically, our baseline includes the following methods: **Base**, which refers to the original model without any edits; **ROME** Meng et al. (2022a), which identifies editing areas using causal mediation analysis framed as a least-squares problem under linear equality constraints and solving it using Lagrange multipliers; **MEND** Mitchell et al. (2022), which employs meta-learning to train a hypernetwork for inferring weight updates from gradients; **MEMIT** Meng et al. (2023), which extends ROME to edit a large set of facts by updating weights in a range of layers; **MeLLo**, which manages multi-hop knowledge editing by decomposing subproblems and detecting conflicts; **PMET**, which optimizes FFN hidden states for precise weight updates, achieving SOTA performance in COUNTERFACT (Meng et al., 2022a) and ZsRE (Levy et al., 2017).

**Setup and Hyperparameters.** To evaluate the performance of different KE methods, we adopt Multi-hop question answering accuracy(Multi-hop Acc) as the primary metric. For each query, the **unedited answer** denotes the expected old fact before knowledge editing, while the **edited answer** represents the expected new fact after editing. Unless otherwise specified, we report the performance of **Base** in generating the **unedited answer** to reflect the original ability of model to leverage knowledge. For the edited model, we report its accuracy in producing the **edited answer**, thereby assessing the effectiveness of the editing method. Our experiments are mainly conducted on the GPT-J (6B) model. We use PMET as our primary experimental method for both the first and furtherance edits and construct a supplementary set from the knowledge triples of MQuAKE-3K to support our **IFMET**. Additional details are presented in Appendix D.2.

| Model | Method | Batch_size=1 | Batch_size=1000 | Batch_size=3000 |
|-------|--------|--------------|-----------------|-----------------|
| | Base | 39.63 | - | - |
| | MeLLo$^\diamond$ Zhong et al. (2023) | 20.3 | 11.0 | 10.2 |
| GPT-J-6B | ROME$^\spadesuit$ Meng et al. (2022a) | 7.6 | - | - |
| | MEMIT$^\spadesuit$ Meng et al. (2023) | 12.3 | 8.1 | 1.8 |
| | MEND$^\spadesuit$ Mitchell et al. (2022) | 11.5 | 4.3 | 3.5 |
| | PMET$^\spadesuit$ Li et al. (2024c) | 11.17 | 11.13 | 11.7 |
| | **IFMET (ours)** | **23.04** | **18.8** | **17.4** |

Table 3: Multi-hop accuracy comparison of different methods on the MQuAKE-3K dataset in a few-shot setting, showing the **Base** model's performance on the unedited answer and the edited model's performance on the edited answer. Methods with $^\spadesuit$ indicate weight-modifying methods, while methods with $^\diamond$ are weight-preserving methods. '-' indicates no relevant result, as ROME does not support multiple edits. **Note:** the **Base** model's performance on the edited answer is 7.70 .

### 5.2 EXPERIMENTAL RESULTS

**General performance.** Table 3 demonstrates the performance of various established methods alongside **IFMET** on MQuAKE-3K. We can easily see the previous weight-modifying approaches generally exhibited poor performance. As the edit batch size increases, all methods except PMET show a certain downward trend. Our method inherits the good batch editing ability of PMET and consis-

tently outperforms all others, showcasing a leading edge. Our approach significantly improves upon existing knowledge editing techniques, demonstrating the effectiveness and necessity of updating knowledge storage in deeper MLP layers. Additionally, we conducted comprehensive ablation study and discussions on generalizability. For detailed results and analyses, please refer to Appendix E.

**Effect of number of hops.** Table 12 in the Appendix displays the performance trends of various knowledge editing methods with different numbers of hops in multi-hop factual recall. We can see that each additional reason hop will negatively impact performance. Notably, IFMET is the best one in all cases, with minimal performance degradation. In particular, its results are close to those of the original model in the two-hop scenarios. This slight decrease underlines IFMET's robustness and its superior ability to handle complex multi-hop tasks effectively.

**Effect of the number of edited instances.** We then consider the performance with different edited instances, which refer to the required number of new knowledge updates in the edit case. The results shown in Appendix Table 11 indicate a performance decline across all methods as more edits are introduced. Notably, IFMET consistently outperforms other approaches, showing the smallest average decline across different instance scenarios. Surprisingly, IFMET achieves an accuracy distribution that is close to that of the original model. It is also the only method that maintains excellent performance in complex two-hop and four-hop scenarios, which even outperforms single-hop cases. This can be more visually observed in Appendix Figure 7.

**Effect of the edit position.** As previously mentioned, the same single-hop fact requires different layers to provide knowledge, depending on its position in the multi-hop reasoning chain, involving the earlier and later MLPs. By categorizing according to position, we can assess whether the editing methods have comprehensively updated the relevant knowledge in the model rather than just making partial updates. We classify the edited case according to its position in the relevant multi-hop reasoning chain as Pre, Mid, and Post. For instance, in a three-hop knowledge sequence, editing the first hop is classified as pre, the second as mid, and the third as post. Please refer to Appendix D.1 for the classification of more complex, multi-edit scenarios. To assess the completeness of our method, we evaluated its performance for both eliminating original knowledge and incorporating new knowledge. As detailed in Table 4, our method significantly enhances outcomes across all classification types—Pre, Mid, and Post. Notably, it achieves exceptional improvements in both modifying new knowledge and eliminating original knowledge, especially in cases classified as Post.

| Editor | Edited Answer ↑ | | | | Unedited Answer ↓ | | | |
|--------|------------------|------|------|------|-------------------|------|------|------|
| | Average Accuracy | Pre | Mid | Post | Average Accuracy | Pre | Mid | Post |
| GPT-J | 7.70 | 6.03 | 16.92 | 7.00 | 39.63 | 38.43 | 35.9 | 44.27 |
| GPT-J+CoT | 6.83 | 5.92 | 9.23 | 7.76 | 42.83 | 41.56 | 39.74 | 47.33 |
| PMET | 11.17 | 12.13 | 16.09 | 6.52 | 29.95 | 23.60 | 35.66 | 41.85 |
| PMET+CoT | 17.04 | 19.84 | 14.32 | 11.91 | 29.35 | 23.12 | **30.43** | 43.22 |
| IFMET | 23.04 | 20.24 | 15.28 | 33.38 | 23.08 | 20.18 | 34.32 | **24.25** |
| **IFMET+CoT** | **31.01** | **31.69** | **19.49** | **35.15** | **21.62** | **17.71** | 30.51 | 26.27 |

Table 4: Multi-hop accuracy comparison between unedited and edited answers using PMET and our editors on the MQuAKE-3K dataset, with edit_batch = 1. The type of edited fact—Pre, Mid, or Post—depends on the edited data position within the multi-hop reasoning chain. Average accuracy is calculated as the weighted average of results from these three categories, which have respective quantities of 1824, 390, and 786. Additionally, +CoT denoted the performance incorporating a Chain-of-thought (CoT) prompt.

## 6 CONCLUSION

We focused on developing locate-then-edit knowledge editing methods for multi-hop factual recall tasks. We first verified that in multi-hop tasks, LLMs tend to retrieve implicit subject knowledge from deeper MLP layers, unlike single-hop tasks, which rely on earlier layers. This distinction explains the poor performance of current methods in multi-hop queries, as they primarily focus on editing shallow layers, leaving deeper layers unchanged. We then proposed IFMET, a novel locate-then-edit KE approach designed to edit both shallow and deep MLP layers. Experimental results demonstrate that IFMET significantly improves performance on multi-hop factual recall tasks.

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

# A RELATED WORK

**Parameter-based Editing** Knowledge editing refers to modifying outdated, inaccurate, or harmful knowledge in LLMs without the need for retraining. Parameter-editing methods achieve this by adjusting the model's internal parameters to update its knowledge while ensuring that information unrelated to the editing domain remains unaffected. An example is ROME (Meng et al., 2022a), which explored the knowledge storage mechanisms in single-hop factual recall tasks based on causal tracing methods and proposed the Rank-One Model Editing method. Together with KN (Dai et al., 2022), it pioneered a paradigm of locate-then-edit, providing guidance for subsequent editing methods. The later extended versions, MEMIT (Meng et al., 2023), MALMEN (Tan et al., 2023), and EMMET (Gupta et al., 2024), further improved ROME by addressing its limitations in large-scale editing, enabling comprehensive edits in a single operation while demonstrating exceptional performance. Meanwhile, PMET (Li et al., 2024c) achieved more precise model editing by decoupling the residual flow of the Transformer into three components: Multi-Head Self-Attention (MHSA), Feed-Forward Networks (FFN), and residual connections, utilizing only the optimized hidden states of the FFN to accurately update FFN weights. Additionally, MEND (Mitchell et al., 2022) trained a hypernetwork to efficiently predict LLM weight updates, enabling rapid knowledge editing. METO (Yin et al., 2024) optimized the model's temporal prediction of facts, editing both historical and new knowledge to reduce forgetting during updates. Wilke (Hu et al., 2024) selected the layers in LLMs that best matched the knowledge pattern for editing, achieving continuous updates and corrections in the model's knowledge. Hewitt et al. (2024) used canonical examples to guide the model editing process, enabling fine-tuned adjustments to model behavior. However, these editing methods primarily focus on knowledge updates in specific layers and lack in-depth optimization for knowledge integration and application in multi-hop reasoning, rendering them inadequate for multi-hop questions. In contrast, IFMET enhances model interpretability, guiding more accurate knowledge integration and thereby improving model performance in multi-hop factual recall tasks.

**Mechanistic Interpretability** LLMs are capable of producing high-quality answers, but their internal workings remain opaque. As a result, the interpretability of LLMs has emerged as both a research hotspot and a critical area of focus. Mechanistic Interpretability refers to the effort to explain the internal mechanisms, decision-making processes, and outputs of LLMs. There are two primary approaches for interpreting large language models (LLMs) in the vocabulary space by examining hidden representations: Probing Classifiers (Belinkov & Glass, 2019; Belinkov, 2022; Wang et al., 2024) and Projecting Representations to the Vocabulary Space (Dar et al., 2022; Merullo et al., 2023; Belrose et al., 2023; Langedijk et al., 2023). The former identifies which parts of the model are crucial for specific tasks by training classifiers, known as probes, on hidden representations, while the latter involves mapping intermediate layer representations to the output vocabulary space and analyzing how these projections predict the next word. In this paper, we focus primarily on Projecting Representations. Logit Lens (nostalgebraist, 2020) extracted outputs corresponding to each layer in the decoding space by applying unembedding operations on the intermediate layers of LLMs. Geva et al. (2022) analyzed the nature of updates at each layer by comparing differences in logit outputs. Merullo et al. (2024) used the Logit Lens to explore how LLMs handle different stages of question-answering tasks. Dar et al. (2022) mapped attention weights of LLMs to lexical space, showing that these weights encode consistent concepts and relations. Belrose et al. (2023) introduced the Tuned Lens, which improves the capability and reliability of the Logit Lens. Finally, Ghandeharioun et al. (2024) proposed the Patchscopes framework, demonstrating that auxiliary models can represent lexical projections through tuning.

Mechanistic Interpretability serves as a tool for debugging and enhancing LLMs and can be applied to a variety of downstream tasks. Xiao et al. (2024) leveraged explanations from multi-head self-attention (MHSA) mechanisms in LLMs by introducing StreamingLLM, a model capable of handling unlimited text without requiring fine-tuning. Through causal tracing, Hendel et al. (2023); Todd et al. (2024) demonstrated that certain attention heads can efficiently encode compact representations of example tasks, leading to improved performance in few-shot prompting. Liu et al. (2024) explored the role of social bias in LLMs, introducing the concept of social bias neurons to explain and mitigate such biases. Furthermore, Li et al. (2024b) proposed an intervention technique during inference, which, based on the interpretability of attention heads, shifts activation values toward "truthful" responses to reduce model hallucinations. In this paper, we analyze the MLP and MHSA components of LLMs to uncover the mechanisms that enable multi-hop reasoning,and building on our findings, we introduce a targeted knowledge-editing method IFMET.

# B MORE DETAILS

## B.1 SUBSET OF MQUAKE

### B.1.1 1-HOP AND 2-HOP SUBSET FOR MECHANISM EXPLORATION

In exploring the mechanisms of fact recall for one-hop and two-hop queries, this experiment utilized cloze templates as the experimental framework. We extracted knowledge from MQuAKE that could answer cloze templates in a zero-shot setting. This approach ensured that the model could recall the knowledge under the strictest conditions while minimizing the impact of unclear responses on the experimental results. The distribution of various relation types across the two subsets is illustrated in Figure 4.

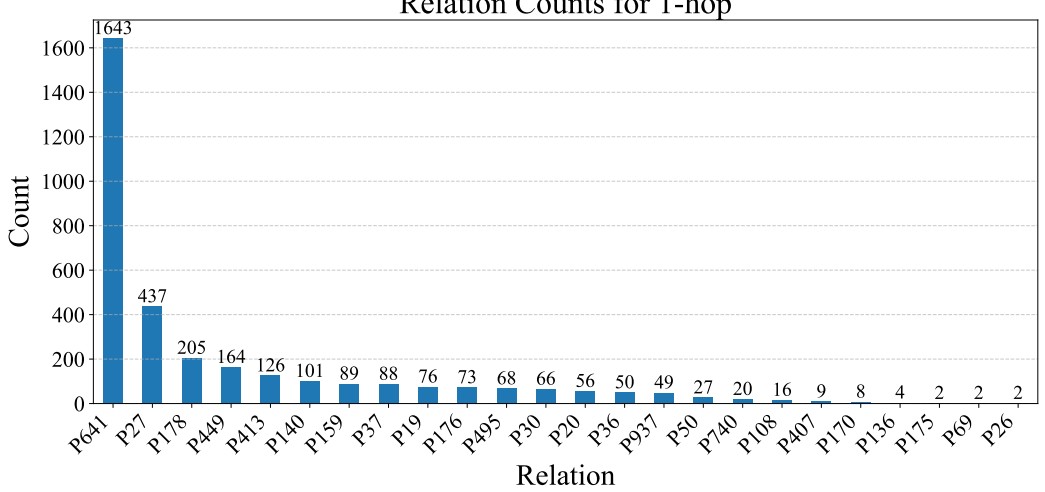

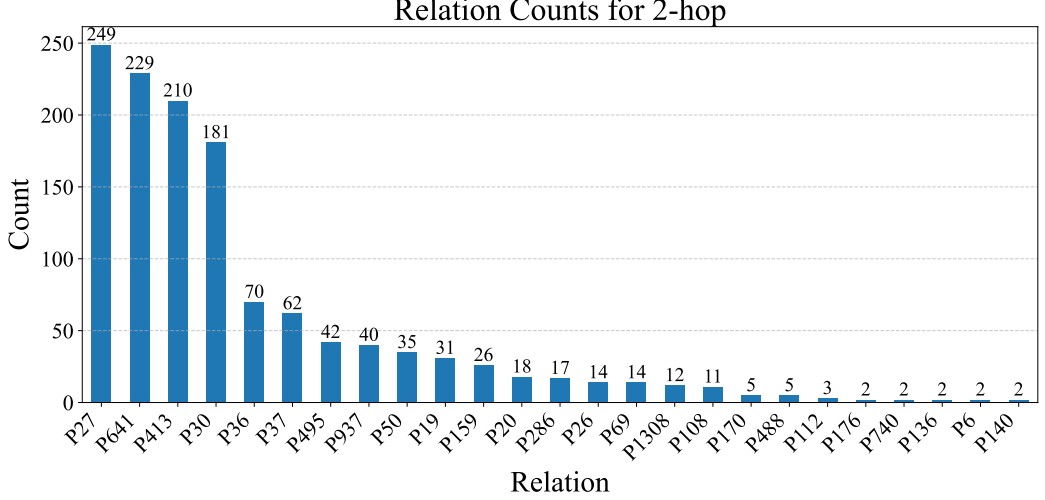

Figure 4: Relation number for 1-hop and 2-hop

### B.1.2 PRE AND POST SUBSET

To construct the subset, we selected two-hop queries from MQuAKE with Cloze-Format templates, and then randomly drew a nearly equal number($\approx 300$) of cases based on the proportion of relations.

## B.2 LEAST SQUARES AND MINIMUM-NORM METHOD

When performing interventions, we need to solve the least squares constraint as follows:

$$\arg\min_{\Delta h_l} \|W_U(h_l + \Delta h_l) - s_l^*\|^2$$

In certain situations, the minimum norm method is more effective than directly solving linear systems or using other numerical methods, especially when the system is underdetermined (i.e., there are fewer equations than unknowns) or when there are infinitely many solutions. The minimum norm method provides a solution with the smallest norm among all possible solutions.

To minimize the probability of the intermediate answer $j$, we replace its logits with the smallest logits of the model's vocabulary, and provide appropriate compensation for the final answer $k$ to maintain the probability of the final answer unchanged. The $\Delta h$ can be represented as:

$$\begin{cases} \Delta h = \Delta h_j + \Delta h_k \\ \Delta h_j = \frac{s_l[j] - s_l^{\min}}{\|W_u[j]\|^2} W_u[j] \\ \Delta h_k = \frac{s_l[k] - s_l^{\min}}{\|W_u[j]\|^2} \alpha W_u[k] \end{cases}$$

The change in the probability of the final answer after causal intervention can be represented by the function $f(\alpha)$: $f(\alpha) = P(h^*, k) - P(h, k)$ Where $f(\alpha)$ is a monotonically increasing function on the interval $(0, 1)$. We can find the zero of this function using the bisection method, ensuring that the final answer, after causal intervention, remains within an acceptable error margin with unchanged probability.

## B.3 CAUSAL INTERVENTION ON SINGLE-HOP CASE

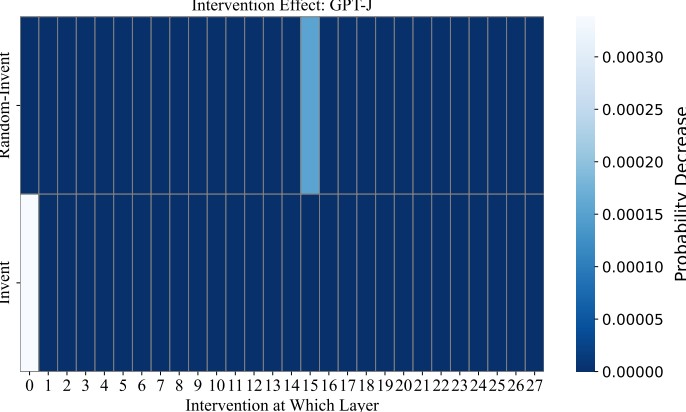

Figure 5: Causal Intervention result of MLP input in last token position in Single-hop case

The results of intervention for single-hop cases are shown in Figure 5. Except for the input layer, no significant effects are shown, indicating that in the single-hop fact recall task, the prediction of the final answer at the last token position is largely independent of the information from the intermediate results.

## C DETAILS OF IFMET

### C.1 DETAILED SUPPLEMENTARY SET CONSTRUCTION PROCESS

We collect 2615 subjects from the MQuAKE dataset. For each subject $s$, we use a Wikidata SPARQL query to retrieve the triplet $(s', r', s)$. The query is illustrated in Table 17. To keep the

query complexity within an acceptable range, we collected all relationships that have appeared in MQuAKE and restricted $r'$ to those that have occurred in the relation set. We then use the prompt 14 to filter out the answerable $(s', r', s)$ triples. For each edit case $(s, r, o \rightarrow o*)$, we are able to construct a two-hop edit template $T_C(s')$ with the multi-hop chain $C = (s', r's) \oplus (s, r, o \rightarrow o*)$.

## C.2 DETAILED EDIT PROCESS

---

**Algorithm 1: IFMET**

---

**Data:** Requested edits $E = \{(s_i, r_i, o_i \rightarrow o_i^*)\}_{i=1}^N$, Supplementary set
$\quad\quad Sup = \{(s_i', r_i', s_i)\}_{i=1}^N$, model $\mathcal{M}$, first edit layers $l_1$, furtherance edit layers $l_2$
**Result:** Modified model $\mathcal{M}_E$ containing edits from $E$

1 **for** $\underline{(s_i, r_i, o_i^*)} \in E$ **do**      // First Edit Process
2     **Generate the single edit prompt** $T_{r_i}(s_i)$ ;
3     **Optimize** $v_i^* \leftarrow Search(T_{r_i}(s_i))$ ;    // $v_i^*$ for every new fact
4 **end**
5 **for** $l \in l_1$ **do**      // Update weights of Shallow MLPs
6     $\Delta^l \leftarrow Calculate([v_1^*, \ldots, v_N^*])$    ;   // Compute weight change with target vectors
7     $W^l \leftarrow W^l + \Delta^l$    ;        // Update layer $l$ MLP weights in model
8 **end**
9 **for** $\underline{(s_i', r_i', s_i)} \in Sup$ **do**      // Furtherance Edit Process
10    **Construct the multi-hop Chain** $C = (s_i', r_i', s_i) \oplus (s_i, r_i, o)$ ;
11    **Generate the multi-hop edit prompt** $T_C(s_i')$ ;
12    **Optimize** $v_i^* \leftarrow Search(T_C(s_i'))$ ;
13 **end**
14 **for** $l \in l_2$ **do**      // Update weights of Deeper MLPs
15    $\Delta^l \leftarrow Calculate([v_1^*, \ldots, v_N^*])$ ;
16    $W^l \leftarrow W^l + \Delta^l$
17 **end**

---

Our method primarily consists of a first edit (step 1-8 in Algorithm 1) and a furtherance edit (step 9-17 in Algorithm 1). Each single edit process obtains target weights via optimizing the objective of knowledge preservation and editing:

$$\underset{\hat{W}}{\arg\min} \left( \lambda \underbrace{\|\hat{W}K_0 - W^{out}K_0\|^2}_{\text{Preserve}} + \underbrace{\|\hat{W}K_E - V_E\|^2}_{\text{Edit}} \right),$$

where $K_0 = \left[ k_0^1 \,|\, k_0^2 \,|\, \cdots \,|\, k_0^N \right]$ and $V_0 = W^{out}K_0$ contain all the knowledge we want to preserve, $K_E = \left[ k_e^1 \,|\, k_e^2 \,|\, \cdots \,|\, k_e^E \right]$ is the matrix containing the edits we try to make and $V_e = [v_{e_1}^* \,|\, \ldots \,|\, v_{e_E}^*]$ represents the target representations of the new knowledge. $(K_E, V_E)$ corresponds to the edited fact set $\{(s_i, r_i, o_i^*) \,|\, i = 1, 2, \cdots, E\}$. We consider the target weight $\hat{W}$ as the sum of the original weight $W^{out}$ and the incremental weight $\Delta$, as explicated in Li et al. (2024c), a closed-form solution to the incremental weight can be derived:

$$\Delta = RK_E^T(C_0 + K_E K_E^T)^{-1}, \quad R \triangleq (V_E - W^{out}K_E), \quad C_0 \triangleq K_0 K_0^T. \tag{3}$$

Thus, solving the optimal parameter $\hat{W}$ is transformed into calculating edited fact representation $\{(k_e^i, v_e^i) \,|\, i = 1, \ldots, E\}$. In this process, an edit instance $e = (s, r, o \rightarrow o*)$, $(k_e, v_e)$ the pre-edited fact $(s, r, o)$ and $(k_e, v_e^*)$ denotes post-edited $(s, r, o*)$. To obtain the target representations of the new knowledge $v_e^* = v_e + \delta$, we optimize the learnable parameter vector $\delta$ to modify the original value vector. $Search$ is the process of obtain the optimized $\delta$ through gradient descent:

$$\delta = \underset{\delta}{\arg\min} \mathcal{L}(\delta) = \mu D_{\text{KL}} \left( P_{\mathcal{M}_e} [t' \,|\, T] \,\|\, P_{\mathcal{M}} [t' \,|\, T] \right) + \varphi \frac{1}{P} \sum_{j=1}^{P} - \log \mathbb{P}_{\mathcal{M}_e} \left[ o^* \,|\, \text{pref}_j \oplus T_e \right],$$

where $T$ is the KL prompt, such as "*s is a* " and $t'$ is the tokens excluding the token for the answer $o^*$, $T_e$ is the prompt for editing, such as "*The capital of Spain is* " , $\varphi$ and $\mu$ serve as the scaling factor for adjusting the loss. $Calculate$ process is using the $v_e^*$ to slove the $\Delta$ which is a function of $v_e^*$. involves substituting the values of $V_e = [v_{e_1}^* \,|\, \ldots \,|\, v_{e_E}^*]$ corresponding to a series of edits into (3) to compute the $\Delta$.

| Method | Stage | Data | Position | Layers |
|--------|-------|------|----------|--------|
| Previous | Only one | Single-hop | Subject Last Token | Shallow |
| IFMET | First Furtherance | Single-hop Two-hop($Sup$) | Subject Last Token Last Token | Shallow Deeper |

Table 5: The main difference between **IFMET** and previous methods. The term **Stage** refers to the phases of the editing process, **Data** denotes the query utilized for editing, **Position** specifies the token position where the editing is applied, and **Layers** indicate the edited layers.

The primary differences between the first and furtherance edits are reflected in the edit prompt $T_e$ and the layers edited. For example, for the edit instance $e = (s, r, o \rightarrow o*)$, the first edit utilized a one-hop edit template $T_e = T_r(s)$ provided by MQuAKE to edit layers [3,8] of the GPT-J model in the subject last token position. For the furtherance edit, a two-hop template $T_e = T_C(s')$ composed of a support case $(s', r, s)$ and $(s, r, o^*)$, and this two-hop template was applied to edit layers [16,20] of the GPT-J model in the last token position.

# D ADDITION EXPERIMENTAL SETTINGS

## D.1 CRITERIA FOR CLASSIFYING DATASET INTO *pre*, *mid*, AND *post*.

Consider a multi-hop question composed of $n$ triples. We define the positions of edits (with index starting from 1) as the set $\{e_1, e_2, \ldots, e_m\}$, where $m$ represents the total number of edits. Edits occurring in $m$ consecutive positions starting from the first hop are classified as *pre* (where $1 \leq m \leq n$), while those occurring from the $(n-m+1)$th to the $n$th position are labeled *post* (also with $1 \leq m \leq n$). Edits not including the first and last hops are categorized as *mid*.

For non-consecutive edits, classification as *pre* or *post* depends on the positions of the first and last hops relative to the edit distance; if the distances are equal, priority is given to *post*. For example, in a three-hop question, an edit at the first hop is classified as *pre*, an edit at the second hop as *mid*, and edits at both the first and second hops are categorized as *pre*.

## D.2 EXPERIMENTAL SETTINGS

When constructing the support set, for each edit case, no more than three supplements per relation were added from the supplementary dataset. The relation types of the supplementary set are the same as MQuAKE. We set the edit batch sizes to 1, 1000, and 3000.

In both the first and furtherance edits, our configuration for PMET adheres to the settings specified by (Li et al., 2024c). Initially, we set $\varphi = 1$ and $0 \leq \mu \leq 1$ to manage the retention of the model's original knowledge. As $\mu$ increases, the retention level also increases, while $\varphi$ exhibits the opposite trend. After maximizing the probability of the target knowledge, we reduce $\varphi$ to 0.1 to preserve the original knowledge as much as possible. Optimization is halted when $D_{\text{KL}} < 0.01$. On GPT-J, for estimating the covariance matrix (i.e., the set of previously memorized keys $C_0$), we sample 10,0000 times on Wikitext in fp32 precision and set $\lambda = 6000$. When optimizing, we limit the total optimization steps to 30 with a learning rate of 0.2. All our experiments were conducted using the MQuAKE dataset. To test the accuracy of answers to multi-hop questions, we adhered to the few-shot in Table 15 and Chain of Thought (CoT) templates in Table 13 and procedures as outlined in (Zhong et al., 2023).

# E ABLATION STUDY AND GENERALIZABILITY OF IFMET

Based on the results of the interpretability analysis, we emphasize the critical role of editing the last token position using the supplementary set and modifying relevant knowledge in the deeper-layer MLPs to enhance multi-hop reasoning accuracy. Given the distinctions between **IFMET** and other existing methods, we highlight four key components, especially in the furtherance edit: two-stage modification, the use of a supplementary set, editing the last token position, and updating

knowledge in the deeper-layer MLPs during the second stage, as illustrated in the table 5. In the following sections, we will focus on analyzing the roles of these key components and attempt to assess the generalizability of the overall method.

## E.1 ABLATION STUDY

| One-Stage Edit | | | Two-Stage Edit | | | Multi-hop Acc | Efficacy |
|---|---|---|---|---|---|---|---|
| **Data** | **Layers** | **Position** | **Data** | **Layers** | **Position** | | |
| Single-hop | Shallow | Subject Last | × | × | × | 11.70 | 94.62 |
| | Deeper | Last | | | | 12.10 | 97.68 |
| Sup | Shallow | Last | | | | 10.00 | 40.45 |
| | Deeper | Subject Last | | | | 9.03 | 13.64 |
| | | Last | | | | 15.50 | 54.15 |
| Single-hop | Shallow | Subject Last | Single-hop | Deeper | Subject Last | 11.85 | 95.21 |
| | | | | | Last | 11.33 | 98.90 |
| | | | Sup | Shallow | Subject Last | 13.97 | 92.41 |
| | | | | | Last | 12.33 | 93.45 |
| | | | | Deeper | Subject Last | 12.90 | 94.69 |
| | | | | | Last | **17.40** | 94.74 |

Table 6: Comparison of different methods across batch sizes, hop numbers, and edit instances.

To examine the performance improvements attributed to the four aforementioned components, we conducted extensive experiments on GPT-J-6B model under the condition of edit_batch = 3000 using the MQUAKE-3K dataset. We add metric **Efficacy** to measure whether an edit has been successfully applied to a model. It is calculated as the percentage of edits where the probability of answer token $P(\text{edited answer}) > P(\text{unedited answer})$ for a given single-hop query prompt used during model editing. The complete results are summarized in the Table 6. For a more intuitive comparison, we have highlighted the contributions of the four components, as shown in Table 7.

Our interpretability analysis has identified that the existing editing methods fail to adequately modify knowledge in the deeper MLP layers, resulting in poor performance on multi-hop factual recall tasks. Additionally, our findings suggest that implicit multi-hop step dependencies rely on the knowledge provided by these deeper MLP layers. Based on these interpretability results at the last token position, we propose the **IFMET**. In the second stage of editing, we use a combination of supplement sets and modifications to the deeper MLP layers to update the knowledge therein.

The three tables 7, 8 and 9, encompass various models and different edit batches, which we believe provide sufficient evidence to substantiate our claims. In all three tables, we have utilized the **PMET** as a baseline to assess method performance. The importance of each component is reflected through comparisons of performance improvements over **PMET**. **PMET**'s performance exemplifies a single-stage edit approach using shallow MLP edits based on single-hop edit query. From the analysis of the ablation experiments, we derive the following conclusions:

- **IFMET**: Firstly, it can be observed that the implementation of **IFMET** achieves the best performance in Multi-hop Acc. In the second stage editing, we employ a multi-hop supplementary set alongside deep MLP editing techniques. Across all the experimental tables mentioned, **IFMET** consistently demonstrates a substantial improvement in inferential performance compared to **PMET**.
- **w/o** $First$: Only modifying the deeper layers using $Sup$ data effectively enhances performance on multi-hop reasoning tasks. However, the absence of first-stage editing results in unchanged knowledge in the earlier layers, leading to poor performance in single-hop fact recall tasks.
- **w/o** $Last$ demonstrated the importance of editing the last token position.
- **w/o** $Sup$: This represents that, in the second editing stage, we continued to use single-hop edit query instead of the supplement set to edit the deeper MLP layers. However, the results corroborate the interpretability analysis which emphasizes the differences between single-hop and multi-hop reasoning mechanisms. Compared to the original one-stage method

| Edits | Editor | Multi-hop Acc | Efficacy |
|---|---|---|---|
| | IFMET | **17.40** (↑**48.7%**) | 94.74 (↑0.1%) |
| | w/o $First$ | 15.50 (↑32.4%) | 54.15 (↓**42.8%**) |
| 3000 | w/o $Sup$ | 11.33 (↓**3.2%**) | 98.90 (↑**4.5%**) |
| | w/o $Last$ | 12.90 (↑10.2%) | 94.69 (↓0.1%) |
| | w/o $Deeper$ | 12.33 (↑5.3%) | 93.45 (↓1.2%) |
| | PMET | 11.70 | 94.62 |

Table 7: The results of the ablation experiments of MQuAKE-3K on GPT-J-6B model. **w/o** $First$ represents only optimizing the deeper MLPs with $Sup$ without modifying shallow MLPs first. **w/o** $Sup$ represents reusing Single-hop rewrite query used in first stage rather than the $Sup$ queries to modify deeper MLPs. **w/o** $Last$ represents second stage editing occured in subject last token position. **w/o** $Deeper$ represents apply second stage editing in shallow MLPs. **PMET** represents the original one-stage method we used. Both the percentages of decrease(↓) and increase(↑) are calculated relative to **PMET** as the baseline. The most significant performance decline is highlighted in **red** and the most significant performance increase is highlighted in **green**.

      **PMET**, performance fluctuations remained within a relatively stable range in contrast to IFMET's own +70% improvement. Therefore, we conclude that using single-hop data combined with deep MLP editing is ineffective, highlighting the critical importance of the supplementary set.

- **w/o** $Deeper$: In this setup, the second-stage editing was modified to use the supplementary set combined with shallow MLP editing(rather than deeper MLP layers). If this also shows a significant performance improvement, it would indicate that merely expanding with the supplementary set, without considering its mechanism on the deeper MLP layers, can enhance results. However, as observed across the three tables, there was a consistent minor fluctuation in performance (ranging from -6.8% to +5.3%). In contrast to **IFMET**'s own +70% improvement, this underscores the importance of editing the deeper MLP layers when using the supplementary set.

In light of the results from the ablation experiments w/o sup and w/o deeper, which align with our interpretability analysis, we emphasize that merely increasing the supplementary set is insufficient. It is essential to apply the supplementary set to the deeper MLP layers for knowledge editing to effectively enhance performance on multi-hop factual recall tasks.

To further investigate whether the **IFMET** method effectively balances the requirements of general knowledge editing and multi-hop fact recall tasks, we constructed the paraphrase set and neighborhood set for a subset of the MQuAKE-CF dataset , following the approach used in the COUNTER-FACT dataset Meng et al. (2022a). We conducted experiments under two configurations: edit_batch = 1 and edit_batch = 100 and evaluate with following additional metrics:

- **Efficacy** measures whether an edit has been successfully applied to a model. It is calculated as the percentage of edits where $P(\text{edited answer}) > P(\text{unedited answer})$ for a given query prompt used during model editing.

- **Paraphrase** evaluates the model's generalization ability under an edit. It is defined as the percentage of edits where $P(\text{edited answer}) > P(\text{unedited answer})$ for paraphrases of the query prompt.

- **Neighborhood** assesses the locality of the model editing, i.e., whether the edit of a specific fact affects other facts stored within the model. Neighborhood score is defined as the percentage of facts in the neighborhood of the edited fact that remain unchanged after the edit.

The results are summarized in the table 8. It can be observed that **IFMET** achieves a significant improvement of over 60% in Multi-hop accuracy and also demonstrates enhancements in both efficacy score and Paraphrase score, at the cost of a minor decrease in the neighborhood score. And only the complete **IFMET** method achieves balanced optimal performance across multiple metrics. Furthermore, we posit that **IFMET**'s performance is closely linked to the one-stage method it

| Edits | Editor | Multi-hop | Efficacy | Specificity | Paraphrase |
|---|---|---|---|---|---|
| | IFMET | 28.38 (↑**78.0%**) | 99.56 (↑12.8%) | 65.06 (↓17.2%) | 90.17 (↑**5.3%**) |
| | w/o $First$ | 23.14 (↑45.1%) | 66.59 (↓**24.6%**) | 59.54 (↓**24.3%**) | 41.48 (↓**51.6%**) |
| | w/o $Sup$ | 17.69 (↑10.9%) | 100.00 (↑**13.3%**) | 77.71 (↓1.2%) | 86.24 (↑0.7%) |
| 1 | w/o $Last$ | 18.12 (↑13.6%) | 88.21 (↑0.0%) | 78.60 (↓**0.0%**) | 86.24 (↑0.7%) |
| | w/o $Deeper$ | 15.07 (↓**5.4%**) | 99.56 (↑12.8%) | 70.31 (↓10.6%) | 86.90 (↑1.5%) |
| | PMET | 15.94 | 88.21 | 78.60 | 85.59 |
| | IFMET | 27.07 (↑**64.8%**) | 96.29 (↑8.1%) | 69.89 (↓9.4%) | 84.28 (↑**3.8%**) |
| | w/o $First$ | 22.71 (↑35.1%) | 73.36 (↓**17.8%**) | 69.21 (↓**10.2%**) | 34.72 (↓**57.3%**) |
| | w/o $Sup$ | 17.25 (↑2.6%) | 99.13 (↑11.3%) | 76.63 (↑**0.6%**) | 84.06 (↑3.5%) |
| 100 | w/o $Last$ | 15.94 (↓**5.2%**) | 89.08 (↓0.0%) | 76.85 (↑0.3%) | 81.55 (↑0.4%) |
| | w/o $Deeper$ | 16.16 (↓3.9%) | 99.56 (↑**11.8%**) | 74.67 (↓3.2%) | 81.00 (↓0.3%) |
| | PMET | 16.81 | 89.08 | 77.07 | 81.22 |

Table 8: The results of the ablation experiments on GPT-J-6B model using a subset of MQuAKE-CF. Both the percentages of decrease(↓) and increase(↑) are calculated relative to **PMET** as the baseline. The most significant performance decline is highlighted in **red** and the most significant performance increase is highlighted in **green**.

builds upon(e.g. **PMET**). Enhancements to the one-stage method are likely to lead to corresponding improvements in **IFMET**'s performance across relevant metrics.

E.2 GENERALIZABILITY OF IFMET

In this subsection, We explore the generalizability of our method from four key perspectives:

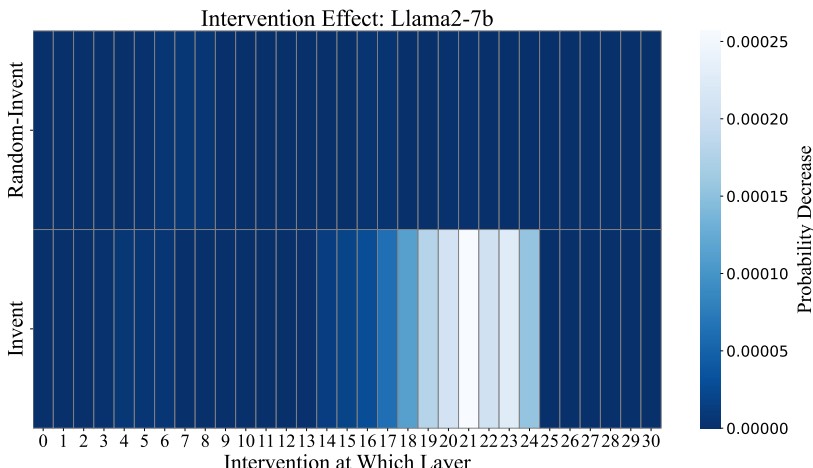

Figure 6: Causal Intervention result of MLP hidden state in last token position on LLaMA-2

**Generalization to other models.** We first extended the causal intervention experiments in Section 3 to the LLaMA-2-7B model. The results shown in Figure 6 demonstrate the consistency of interpretability analysis across models, demonstrating the critical role of deeper-layer MLPs in LLaMA-2 model for multi-hop fact recall tasks. Additionally, we repeated the ablation experiments on LLaMA-2-7B to evaluate the generalizability of **IFMET**. The results shown in Table 9 are consistent with those observed on GPT-J, highlighting the importance of the four components in **IFMET** as well as the superiority of the method itself on LLaMA-2-7B model. Considering both the interpretability analysis and experimental outcomes, we conclude that our analysis and method are equally applicable to larger models, such as LLaMA-2.

**Construction of the supplementary set.** In **IFMET**, we emphasize the importance of constructing multi-hop reasoning supplementary set. In this work, we collect this supplementary set leveraging WikiData and SPARQL. However, it is important to note that any other valid knowledge base can

| Edits | Editor | Multi-hop | Efficacy | Specificity | Paraphrase |
|---|---|---|---|---|---|
| 1 | IFMET | 28.38 (↑**73.3%**) | 99.78 (↑**13.7%**) | 65.50 (↓10.8%) | 75.00 (↑**23.1%**) |
| | w/o $First$ | 25.76 (↑57.3%) | 56.55 (↓**35.6%**) | 62.18 (↓**15.3%**) | 39.08 (↓**35.9%**) |
| | w/o $Sup$ | 19.43 (↑18.6%) | 98.68 (↑12.4%) | 70.35 (↓4.2%) | 69.00 (↑13.2%) |
| | w/o $Last$ | 15.72 (↓4.1%) | 88.86 (↑1.2%) | 73.03 (↓**0.6%**) | 61.90 (↑1.6%) |
| | w/o $Deeper$ | 15.28 (↓**6.8%**) | 96.72 (↑10.1%) | 65.61 (↓10.7%) | 66.99 (↑9.9%) |
| | w $Sup_{model}$ | 26.86 (↑64.0%) | 96.07 (↑9.4%) | 63.32 (↓13.8%) | 74.24 (↑21.8%) |
| | PMET | 16.38 | 87.77 | 73.41 | 60.92 |
| 100 | IFMET | 27.29 (↑**76.0%**) | 97.82 (↑4.2%) | 65.50 (↓9.9%) | 84.17 (↑**14.2%**) |
| | w/o $First$ | 24.67 (↑59.2%) | 64.41 (↓**31.4%**) | 63.97 (↓12.0%) | 41.81 (↓**43.3%**) |
| | w/o $Sup$ | 15.07 (↓2.8%) | 99.34 (↑**5.8%**) | 71.83 (↓1.2%) | 76.64 (↑4.0%) |
| | w/o $Last$ | 13.97 (↓**9.9%**) | 94.32 (↑0.4%) | 72.14 (↓**0.8%**) | 74.67 (↑1.3%) |
| | w/o $Deeper$ | 15.94 (↑2.8%) | 96.51 (↑2.8%) | 69.48 (↓4.4%) | 75.44 (↑2.3%) |
| | w $Sup_{model}$ | 22.49 (↑45.1%) | 96.07 (↑2.3%) | 63.32 (↓**12.9%**) | 79.26 (↑7.5%) |
| | PMET | 15.50 | 93.89 | 72.66 | 73.69 |

Table 9: The results of the ablation experiments on LLaMA-2-7B model using a subset of MQuAKE-CF. **w** $Sup_{model}$ represents the $Sup$ queries generated by model itself to modify deeper MLPs. Both the percentages of decrease(↓) and increase(↑) are calculated relative to **IFMET** as the baseline.

| Model | Method | Time |
|---|---|---|
| GPT-J-6B | MEMIT | 4.5s |
| | PMET | 5.0s |
| | IFMET | 9.7s |
| LLaMA-2-7B | MEMIT | 2.1s |
| | PMET | 2.0s |
| | IFMET | 3.4s |

Table 10: The average time required to edit a single case varies across methods. For the two one-stage methods, **MEMIT** and **PMET**, this corresponds to the process of optimizing the shallow-layer MLPs using single-hop queries. For **IFMET**, the process includes updating the deeper-layer MLPs using two-hop supplementary sets.

replace WikiData and SPARQL. A straightforward alternative is to treat the model itself as a reliable knowledge base for extracting relevant knowledge.

To test this hypothesis, we used a simple prompt to retrieve relevant knowledge directly from the model for constructing the supplementary set, as illustrated in the example prompt 16. Due to computational and time constraints, we limited each case to a minimum of one supplementary entry and a maximum of five supplementary entries. The results of substituting the original supplementary set with one generated by LLaMA-2 itself for editing are also shown in Table 9 called **w** $Sup_{model}$. The results show a significant improvement over the one-stage **PMET**, with performance trends aligning closely with those of **IFMET**. Notably, minimal effort was invested in designing the knowledge retrieval prompt, and no additional filtering or preprocessing was applied. This suggests that the supplementary set generated by the model represents a relatively low-quality version, effectively serving as a lower bound for the method's performance across various metrics. Despite this, it still outperforms existing one-stage methods. This highlights the inherent superiority of the **IFMET** framework and demonstrates the feasibility of using the model itself to construct the supplementary set.

**Time complexity of IFMET.** We compared the time complexity of **IFMET** with that of the one-stage **PMET** method it builds upon, the result is shown in Table 10. On average, the time required to perform a complete edit for a single case on GPT-J using **IFMET**(with supplementary set) was approximately 2.5× that of **PMET**. For LLaMA-2, the time required was about 1.5× that of **PMET**. We believe this is within an acceptable range, and as the editing speed of the single-stage method improves, the **IFMET** framework will correspondingly become faster.

**Comparison with Weight-Preserving Methods** Although there have been some weight-preserving editing methods(e.g. RAG-based Methods) accessing good performance for multi-hop question answering in KE scenario, we believe that exploring the locate-then-edit methodology remains meaningful for several reasons:

1. **From the perspective of understanding internal knowledge utilization**: The mechanisms underlying a model's use of internal knowledge differ fundamentally from those governing the use of external knowledge Jin et al. (2024). Investigating the potential of locate-then-edit methods holds significant value for advancing the interpretability of internal knowledge processes, laying the groundwork for deeper insights and practical implementations. Additionally, we believe this approach enables a more fundamental and precise modification of knowledge.

2. **From a practical standpoint**: Methods based on retrieval-augmented generation (RAG) require providing extensive contextual input tokens, posing substantial challenges in terms of computational efficiency and hardware demands. And these methods face several challenges. Instead of injecting knowledge into LLMs, they retrieve related facts stored in memory for editing. As a result, their retrieval success rates become crucial, particularly when managing complex real-world scenarios involving exponential growth in knowledge updates. Moreover, we argue that an over-reliance on modifying knowledge through external contexts introduces security risks, as it may be exploited for data theft and attacks Upadhayay et al. (2024), especially in real-world applications.

# F ADDITIONAL EXPERIMENTAL RESULTS

| Model | Method | Average Accuracy | 1-Edit | 2-Edit | 3-Edit | 4-Edit |
|-------|--------|------------------|--------|--------|--------|--------|
| | Base | 42.83 | 36.96 | 45.27 | 46.85 | 48.51 |
| GPT-J-6B | FT | 1.9 | 4.2 | 0.7 | 0.3 | 0.0 |
| | MEND | 11.5 | 16.0 | 11.0 | 7.3 | 4.4 |
| | ROME | 18.1 | 23.8 | 20.9 | 9.0 | 2.6 |
| | MEMIT | 12.3 | 20.5 | 9.8 | 5.5 | 2.6 |
| | PMET | 17.04 | 22.63 | 16.74 | 11.19 | 7.84 |
| | **IFMET (ours)** | **31.01** | **30.26** | **35.21** | **24.30** | **31.72** |

Table 11: **Multi-hop Acc** Performance comparing the baselineand our method with CoT on multi-hop questions in MQuAKE-3k, categorized by the number of edits 1, 2, 3, 4. **Base** in this table represents unmodified GPT-J-6B model, and we report its performance on **unedited answer** with CoT.

| Model | Method | Average Accuracy | 2-hop | 3-hop | 4-hop |
|-------|--------|------------------|-------|-------|-------|
| | Base | 42.83 | 48.9 | 30.7 | 48.9 |
| GPT-J-6B | FT | 1.9 | 3.7 | 1.4 | 0.5 |
| | MEND | 11.5 | 13.9 | 11.3 | 9.5 |
| | ROME | 18.1 | 33.8 | 9.1 | 11.4 |
| | MEMIT | 12.3 | 22.5 | 6.0 | 8.4 |
| | PMET | 17.04 | 26.65 | 12.76 | 11.7 |
| | **IFMET (ours)** | **31.01** | **44.06** | **23.58** | **25.4** |

Table 12: **Multi-hop Acc** Performance comparing the baseline and our method with CoT on multi-hop questions in MQuAKE-3k, categorized by hop counts of 2, 3, 4. **Base** in this table represents unmodified GPT-J-6B model, and we report its performance on **unedited answer** with CoT.

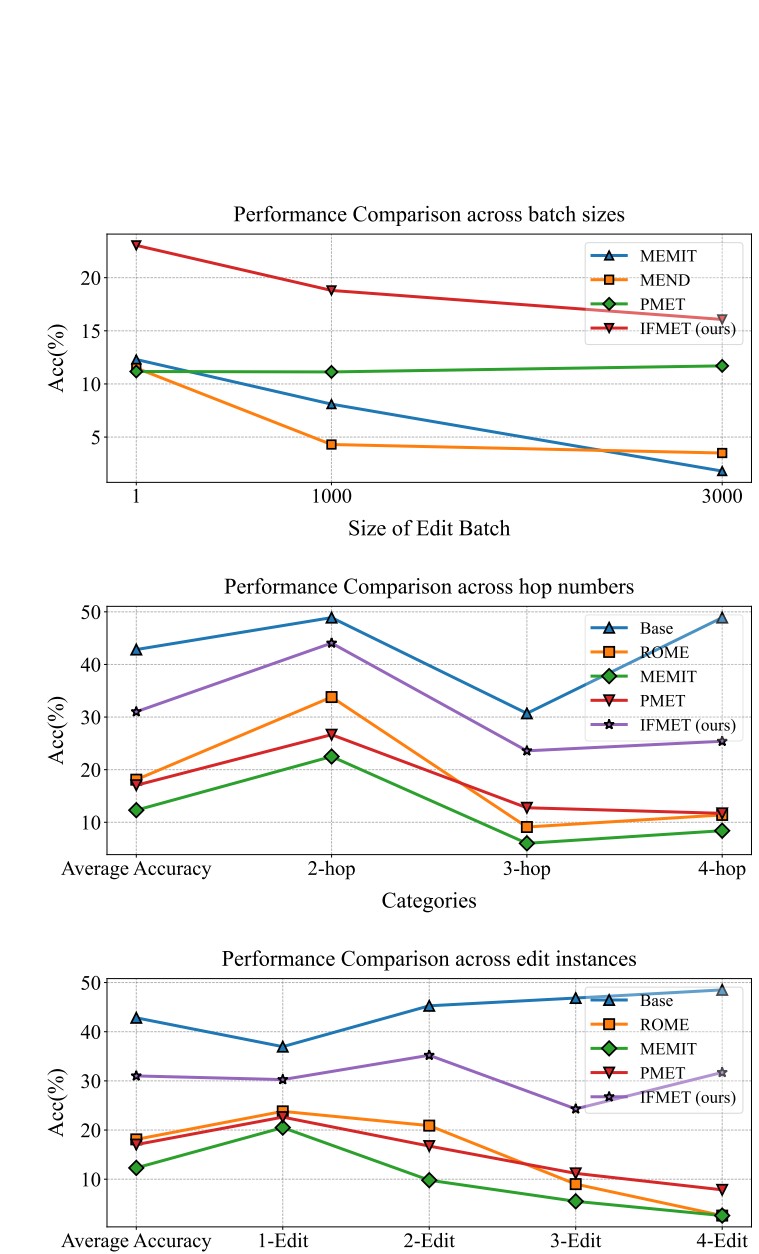

Figure 7: **Multi-hop Acc** Performance Comparison of different methods across batch sizes, hop numbers, and edit instances. **Base** in this table represents unmodified GPT-J-6B model, and we report its performance on **unedited answer** with CoT.

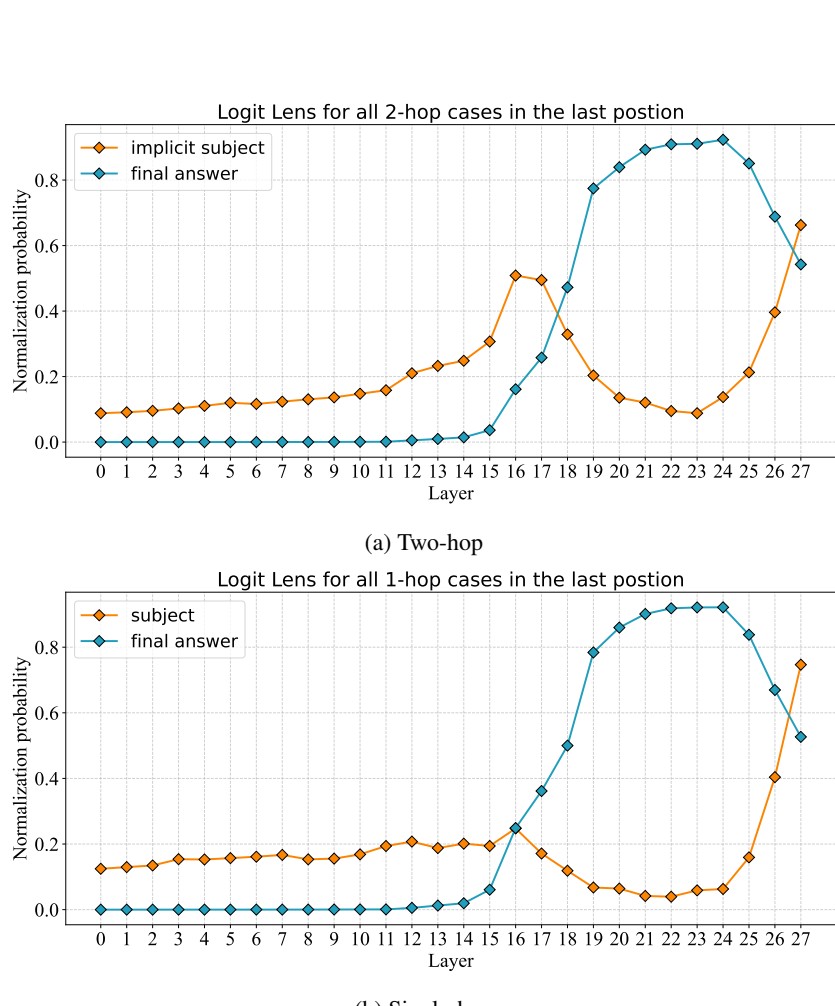

(a) Two-hop

(b) Single-hop

Figure 8: **LogitLens results of the last token position at different layers.** (a) Yellow line represents the information containing implicit subject $s_2$, i.e., Info$(h_l, s_2)$. Blue line represents the information for the final answer, i.e., Info$(h_l, o_2)$. (b) Yellow line represents the information of subject $s$. i.e., Info$(h_l, s)$ and Blue line represents the information of the answer $o$, i.e., Info$(h_l, o)$. Larger versions of the sub-figures are available in the Appendix

```
Question:  What is the capital of the country where Plainfield Town Hall
is located?
Thoughts:  Plainfield Town Hall is located in the country of the United
States of America.  The capital of United States is Washington, D.C.
Answer:  Washington, D.C.

Question:  In which country is the company that created Nissan 200SX
located?
Thoughts:  Nissan 200SX was created by Nissan.  Nissan is located in the
country of Japan.
Answer:  Japan

[3 in-context demonstrations abbreviated]

Question:  Who has ownership of the developer of the Chevrolet Corvette
(C4)?
Thoughts:  The developer of Chevrolet Corvette (C4) is Chevrolet.
Chevrolet is owned by General Motors.
Answer:  Model Generated Answer Goes Here
```

Table 13: The template of the prompt we used for asking multi-hop questions using chain-of-thoughts.

```
(In-context-learning examples)
Q: Who is the developer of Telegram?  A: Telegram FZ-LLC
Q: Who is the developer of Microsoft Windows?  A: Microsoft
Q: Who is the developer of PlayStation 2?  A: Sony Interactive
Entertainment
Q: Who is the developer of iTunes?  A: Apple Inc.
Q: Who is the developer of SR-71 Blackbird?  A: Kelly Johnson
Q: Who is the developer of Moblin?  A: Linux Foundation
Q: Who is the developer of Xbox 360?  A: Microsoft
Q: Who is the developer of Kinsey scale?  A: Alfred Kinsey
(Query during inference)
Q: Who is the developer of SteamOS? A:Valve Corporation
```

Table 14: An example of the prompt we used to recall single-hop fact

```
(In-context-learning examples)
Q: What is the country where The Rotunda is located?  A: United States of
America
Q: In which country was Tohar Butbul granted citizenship?  A: Israel
Q: Who was Nissan 200SX created by?  A: Nissan
Q: What continent is the country where Prickly Pear grows located in?  A:
Europe
Q: What is the capital of the country where Plainfield Town Hall is
located?  A: Washington, D.C.
Q: In which country is the company that created Nissan 200SX located?  A:
Japan
Q: Who was Dodge Ram SRT-10 created by?  Dodge
Q: Who is the spouse of Joe Biden?  A: Jill Biden
Q: Which continent is the country where the director of "My House
Husband:  Ikaw Na!" was educated located in?  A: Asia
Q: What country was the location of the Battle of Pressburg?  A: Hungary
Q: Who is the spouse of the US president?  A: Jill Biden
Q: Who has ownership of the developer of the Chevrolet Corvette (C4)?  A:
General Motors
Q: Who is Joe Biden married to?  A: Jill Biden
Q: What is the country of citizenship of Charles II of Spain?  A: Spain
Q: Who was Chevrolet Biscayne created by?  A: Chevrolet
Q: What is the name of the current head of state in United Kingdom?  A:
Elizabeth II
Q: multi-hop question
```

Table 15: The template of the prompt we used for asking multi-hop questions using few shot.

```
(In-context-learning examples)
Input:  The country that has nationals <mask> is located in the continent
of Asia
Output:  Hitomi Yaida
Input:  The country that has nationals <mask> has the official language
of Italian
Output:  Giorgio Chiellini
Input:  The university where <mask> was educated located its headquarters
in the city of Vienna
Output:  Michael Haneke
Input:  The country that has nationals <mask>, its capital is Washington
Output:  Lou Pearlman
Input:  The person who found <mask> is a citizen of United States of
America
Outout:  Microsoft
Input:  The creator of <mask> hails from Italy
Output:  Ferrari
Input:  The author of <mask> is a citizen of United States of America
Output:  Holly Potter
Input:  The person who discovered <mask> lives in Germany
Output:  Volkswagen
Input:  question
```

Table 16: The template of the prompt we used for asking LLaMA-2-7B to generate the supplementary set.

```
SELECT ?subject ?subjectLabel ?predicate ?predicateLabel
WHERE
?subject ?predicate wd:ss.
FILTER (?predicate IN (wdt:relation))
SERVICE wikibase:label  bd:serviceParam wikibase:language
"en".
LIMIT 50
```

Table 17: The template of the SPARQL Query we used for the supplementary triplets.