# OpenReview forum: "Locate-then-edit for Multi-hop Factual Recall under Knowledge Editing"
_ICLR.cc/2025/Conference — Submitted to ICLR 2025_

### Official Review · Reviewer_ocTZ · 2024-11-03

**Soundness:** 2
**Presentation:** 2
**Contribution:** 3
**Rating:** 6
**Confidence:** 4

**Summary:**

The paper introduces IFMET, a novel locate-then-edit approach for knowledge editing in Large Language Models (LLMs), specifically targeting multi-hop factual recall tasks. The authors leverage mechanistic interpretability to identify that LLMs retrieve implicit subject knowledge from deeper MLP layers in multi-hop tasks, unlike single-hop tasks, which rely on earlier layers. This insight leads to the development of IFMET, which edits both shallow and deep MLP layers. Experimental results show that IFMET significantly outperforms existing methods on multi-hop factual recall tasks.

**Strengths:**

1. The paper provides a thorough and interesting analysis of the differences between single-hop and multi-hop tasks, highlighting the importance of editing deeper layers in LLMs.

2. The writing is clear, making the proposed method easy to understand and follow.

3. The experimental results are impressive, demonstrating significant improvements over existing methods in multi-hop factual recall tasks.

**Weaknesses:**

1. The method may have certain limitations. Specifically, using WikiData and SPARQL queries to construct the supplementary dataset may not be applicable to non-factual or non-encyclopedic data, limiting the generalizability of the approach.

2. The organization of the paper could be improved. The experimental section is relatively short, with some important experimental tables relegated to the appendix. Additionally, the paper lacks a comprehensive ablation study.

3. The explanation and analysis of the experimental results are somewhat lacking. For instance, in Table 3, it is unclear what the values represent (e.g., accuracy) and why the performance of IFMET improves as the edit batch size increases, while the performance of other baselines decreases. More detailed explanations and discussions of these observations would strengthen the paper.

**Questions:**

Please refer to the Weaknesses.

---

> ### Author Response · Authors · 2024-11-22
>
> Thank you for your positive feedback on our paper and for your valuable suggestions. Please find our detailed responses to the weaknesses and questions below.
> > W1：The method may have certain limitations. Specifically, using WikiData and SPARQL queries to construct the supplementary dataset may not be applicable to non-factual or non-encyclopedic data, limiting the generalizability of the approach.
>
> In fact,  we introduced a general method for constructing the supplementary set using the model itself and reported its performance in Table 9(w $Sup_{model}$). See **Appendix E.2 for details.** Given that we employed the simplest version of the knowledge retrieval prompt without any post-processing, we consider this to represent the lower bound of performance for this alternative supplementary set construction method. While its performance is slightly lower compared to the version implemented with WikiData and SPARQL, it still exceeds the performance of the original one-stage PMET by over 50%. Therefore, we believe that the method of constructing the supplementary set is highly scalable and generalizable.
>
> > W2：The organization of the paper could be improved. The experimental section is relatively short, with some important experimental tables relegated to the appendix. Additionally, the paper lacks a comprehensive ablation study.
>
> We have added a comprehensive ablation study and generalizability analysis in our new section  \textbf{Appendix E: ABLATION STUDY AND GENERALIZABILITY OF IFMET}. These results demonstrate the positive contributions of the supplementary set and the interpretability-guided targeting of later MLP layers and the last token position. And we discuss the generalizability of our method in many perspectives. Please refer to the corresponding section in the updated PDF for further details.
>
> Regarding the organization of the paper, we fully understand the difficulties you encountered when important results were placed in the appendix. We spent considerable time attempting to incorporate more tables into the main text but were ultimately unsuccessful. At the same time, we believe that the analytical section preceding the experiments is necessary. Therefore, we kindly ask for your understanding that due to space constraints, we had to place some tables, and even the entire ablation study, in the appendix. Our aim was to preserve the completeness of the experimental results as much as possible. All changes are highlighted in blue in the revised PDF file. We hope these revisions address your concerns comprehensively.
>
> > W3: The explanation and analysis of the experimental results are somewhat lacking. For instance, in Table 3, it is unclear what the values represent (e.g., accuracy) and why the performance of IFMET improves as the edit batch size increases, while the performance of other baselines decreases. More detailed explanations and discussions of these observations would strengthen the paper.
>
> We reported the model's performence of Multihop question answering accuracy. To make our paper more comprehensive and clearer, in the revised version we newly provided detailed explanations of the reported outcomes in tables of Section 5.1 **Setup and Hyperparameters** and **Experimental Setup**, ensuring consistent definitions for **"unedited answer"** and **"edited answer"**. Additionally, we have included annotations below each table to specify the exact names of the reported results, along with the models and datasets used. We have also provided detailed explanations of the metrics, such as **neighbour score** and **paraphrase score**, mentioned in the ablation experiments within the text. As shown in Table 3, the editing performance of PMET remains relatively stable across different batch sizes, which we attribute to the inherent superiority of the PMET method[1]. Notably, IFMET, which leverages PMET as the primary method for its first-stage and furtherance editing, inherits this advantage to some extent, demonstrating minimal performance degradation when the batch size changes. As shown in Table 11, IFMET demonstrates strong multi-hop reasoning accuracy when editing four pieces of knowledge within a single reasoning chain. We attribute this to the effectiveness of the two-stage editing process. By correctly updating the knowledge associated with the implicit reasoning steps in the deep-layer MLPs, IFMET achieves a unique capability compared to previous methods. Unlike previous approaches, where performance often deteriorates as the number of edits in a reasoning chain increases, IFMET exhibits improved performance as the number of edits grows within a single reasoning chain.
>
> [1] PMET: Precise Model Editing in a Transformer. AAAI 2024

---

> ### Author Response · Authors · 2024-11-23
>
> We are eager to know if you have any additional concerns or suggestions regarding our paper. If there are none, we sincerely hope you might consider raising the review score, and we would be more than willing to engage in detailed technical communication to address any potential concerns.

---

> ### Author Response · Authors · 2024-11-25
> **Request for Feedback Before Rebuttal Deadline**
>
> As we approach the rebuttal discussion deadline, we would like to follow up with you regarding the additional ablation experiments, generalization analysis, and writing updates we provided. Have these addressed your concerns? If so, we kindly ask if you would consider revising your review score accordingly.
> Thank you for your time and thoughtful evaluation of our work.

---

> ### Comment · Reviewer_ocTZ · 2024-11-25
>
> Thank you for the author's response. The reply has addressed most of my concerns, and therefore, I will maintain my positive score.

---

### Official Review · Reviewer_bAUb · 2024-11-03

**Soundness:** 4
**Presentation:** 3
**Contribution:** 3
**Rating:** 8
**Confidence:** 4

**Summary:**

This paper addresses an important limitation in knowledge editing for LLMs - their poor performance on multi-hop factual recall tasks after editing. Through careful mechanistic analysis using tools like LogitLens and causal intervention experiments, the authors discover that LLMs process multi-hop queries differently from single-hop ones - they accumulate implicit subject information in middle layers before retrieving the final answer from deeper MLP layers. This explains why current KE methods, which focus on editing shallow layers, fail at multi-hop tasks. Based on these insights, they propose IFMET, a novel locate-then-edit approach that edits both shallow and deep MLP layers using supplementary knowledge sets and multi-hop prompts. Their experimental results on the MQuAKE dataset show improvements over existing methods, particularly for complex multi-hop queries.

**Strengths:**

1. Strong Empirical Analysis: The authors provide a thorough, well-designed investigation into how LLMs process multi-hop queries differently from single-hop ones. The use of LogitLens and causal intervention experiments to track information flow through different layers is particularly impressive.

2. Novel Theoretical Insight: The discovery that implicit subject information accumulates in middle layers before being used to retrieve answers from deeper MLP layers is an important contribution to our understanding of LLM reasoning. This finding not only explains the limitations of current KE methods but also has broader implications for LLM interpretability research.

3. Good result: The proposed IFMET solution is elegant and well-motivated by the theoretical findings. This is a practical solution that improves performance on multi-hop queries.

**Weaknesses:**

Overall, this paper is solid. But some parts of this paper are missing:

1. Why do we need to conduct "Locate-then-edit" editing when we have some very good performance RAG-based editing methods [1][2]? There is no need for this paper to compare performance with RAG-based editing in the experiments part, but a short discussion should be included on why IFMET is superior to these RAG-based methods.

2. Currently, only experiments are conducted on MQUAKE-3k; more benchmarks, such as MQUAKE-T, could be included.

[1] Shi, Yucheng, et al. "Retrieval-enhanced Knowledge Editing in Language Models for Multi-Hop Question Answering." CIKM, 2024.
[2] Gu, Hengrui, et al. "Pokemqa: Programmable knowledge editing for multi-hop question answering." ACL, 2024.

**Questions:**

See weakness

---

> ### Author Response · Authors · 2024-11-22
>
> Thank you for your positive feedback on our paper and for your valuable suggestions. Please find our detailed responses to the weaknesses and questions below.
>
> > W1: Why do we need to conduct "Locate-then-edit" editing when we have some very good performance RAG-based editing methods? There is no need for this paper to compare performance with RAG-based editing in the experiments part, but a short discussion should be included on why IFMET is superior to these RAG-based methods.
>
> This is a very interesting question, and to address it, we have added a dedicated section **Comparison with Weight-Preserving Methods** in the new section **Appendix E**. Please review the updated PDF to verify these changes.
>
> > W2：Currently, only experiments are conducted on MQUAKE-3k; more benchmarks, such as MQUAKE-T, could be included.
>
> Due to the computational constraint, we have not tested IFMET on additional benchmarks. However, given the generalizability of PMET, we believe that IFMET will exhibit consistent performance across different settings. Additionally, we have added  extensive ablation studies and generalizability experiments in the new section **Appendix E: ABLATION STUDY AND GENERALIZABILITY OF IFMET**, aiming to further validate the effectiveness of our method.

---

> ### Comment · Reviewer_bAUb · 2024-11-25
>
> Thank you for your rebuttal. I have no further questions.

---

### Official Review · Reviewer_1m6u · 2024-11-04

**Soundness:** 3
**Presentation:** 3
**Contribution:** 3
**Rating:** 5
**Confidence:** 4

**Summary:**

This paper introduces IFMET, a novel locate-then-edit method for knowledge editing for language models.
It aims to improve the multi-hop factual recall by modifying both shallow and deep layers of model weights.
Previous methods focus on shallow layers and thus struggle with multi-hop editing.
IFMET addresses this by using multi-hop prompts and supplementary sets to locate and edit knowledge across multiple layers.
This allows the language model to answer multi-hop queries more accurately post-edit.
Experimental results show that IFMET outperforms existing knowledge editing methods (such as Mello, and MEMIT) on multi-hop knowledge editing.

**Strengths:**

1. This paper fills a gap in KE for multi-hop knowledge editing. It addresses the limitations of traditional local-then-edit methods that primarily operate in shallow layers.
2. The paper gives detailed explanations and experiments about how single-hop and multi-hop queries are handled differently within language models. The used LogitsLen and causal intervention experiments are interesting.
3. The proposed IFMET is compared with strong baselines across experiments and shows great performance.

**Weaknesses:**

1. The used language model GPT-J (6B) is a bit outdated. More recent language models are expected to be considered.
2. How about the performance of the generality and locality of knowledge editing? The editing should also affect related facts but not affect other unrelated facts.
3. The paper lacks discussions about editing efficiency. What is the time complexity of IFMET?
4. Compared to Mello, IFMET requires supplementary set construction from external sources (same for MEMIT and ROME). This may hinder its applications in real cases.

**Questions:**

1. The biblology style should be consistent. Some lines are underlined and others are not.
2. How does IFMET perform on other language models? Like recent Llama-3.
3. How long does IFMET take to edit one multi-hop sample?
4. Can IFMET deal with unstructured facts for knowledge editing as in [1]?
5. Line 387, cizizenship --> citizenship.
6. Figure 1, Mardrid --> Madrid.

 [1] Updating Language Models with Unstructured Facts: Towards Practical Knowledge Editing

---

> ### Author Response · Authors · 2024-11-22
>
> Thank you for your positive feedback on our paper and for your valuable suggestions. Please find our detailed responses to the weaknesses and questions below.
> > Q1:The biblology style should be consistent. Some lines are underlined and others are not.
>
> We have standardized the formatting of the references in new version, removing the underlining annotations.
>
> > Q2&W1: How does IFMET perform on other language models? Like recent Llama-3.
>
> We first extended the causal intervention experiments to the LLaMA-2-7B model. The result demonstrate the consistency of interpretability analysis across models, demonstrating the critical role of deeper-layer MLPs in LLaMA-2 model for multi-hop fact recall tasks. Additionally, we repeated the ablation experiments on LLaMA-2-7B to evaluate the generalizability of **IFMET**. The results shown in table 9 (in new PDF) are consistent with those observed on GPT-J, highlighting the superiority of the method **IFMET** on LLaMA-2-7B model. Considering both the interpretability analysis and experimental outcomes, we conclude that our analysis and method are equally applicable to newer models, such as LLaMA-2. You can refer to this section Generalization to other models in  **Appendix E.2: GENERALIZABILITY OF IFMET** in new PDF for more detailed ablation experiment results.
>
> > W2: How about the performance of the generality and locality of knowledge editing? The editing should also affect related facts but not affect other unrelated facts.
>
>
> To explore this, We constructed the paraphrase set and neighborhood set for a subset of the MQuAKE-CF dataset, following the approach used in the COUNTERFACT dataset. The results indicate that, Compared to the one-stage PMET, IFMET achieves a significant improvement of over **60%** in Multi-hop accuracy and also demonstrates enhancements in both efficacy score and Paraphrase score, at the cost of a minor decrease in the neighborhood score. So that IFMET method achieves balanced optimal performance across multiple metrics. For more details on related ablation studies, please refer to the  section **Appendix: E ABLATION STUDY AND GENERALIZABILITY OF IFMET**.
>
> > W3 & Q3: The paper lacks discussions about editing efficiency. What is the time complexity of IFMET?
>
> We compared the time complexity of IFMET with that of the one-stage PMET method it builds upon. On average,the time required to perform a complete edit for a single case on GPT-J using IFMET(with supplementary set) was approximately 2.5x that of PMET. For LLaMA-2, the time required was about 1.5x that of PMET. We believe this is within an acceptable range, and as the editing speed of the single-stage method improves, the IFMET framework will correspondingly become faster. See **Appendix E.2 for details.**
>
> > W4: Compared to Mello, IFMET requires supplementary set construction from external sources (same for MEMIT and ROME). This may hinder its applications in real cases.
>
> In fact,  we introduced a general method for constructing the supplementary set using the model itself and reported its performance in Table 9(w $Sup_{model}$). See **Appendix E.2 for details.** Given that we employed the simplest version of the knowledge retrieval prompt without any post-processing, we consider this to represent the lower bound of performance for this alternative supplementary set construction method. While its performance is slightly lower compared to the version implemented with WikiData and SPARQL, it still exceeds the performance of the original one-stage PMET by over 50%.
>
> > Q5 & Q6: Line 387, cizizenship --> citizenship. Figure 1, Mardrid --> Madrid.
>
> We have fixed the typo.
>
> > Q4: Can IFMET deal with unstructured facts for knowledge editing?
>
> The application of IFMET to unstructured knowledge editing presents a compelling avenue for further research. Unstructured text inherently encompasses more complex semantic information, and in scenarios involving longer and more intricate texts, the presence of noise can significantly complicate the reasoning process. At present, combining \textbf{IFMET} with efficient fact triple extraction techniques appears to be a feasible approach. We believe this context warrants deeper investigation in future work.
>
> All new changes are highlighted in blue in the revised PDF file and we have included an entire section in the appendix dedicated to ablation experiments and discussions on generalizability. We hope these revisions address your concerns comprehensively.

---

> > ### Comment · Reviewer_1m6u · 2024-11-26
> > **Response to authors**
> >
> > Thank the authors for the response, but some of my concerns weren't addressed, like the performance on newer models. So I want to keep my score.

---

> > > ### Author Response · Authors · 2024-11-26
> > >
> > > Dear Reviewer 1m6u,  In the newly added Table 9, we report performance and ablation experiments conducted on Llama-2-7B. We believe that Llama2-7B, to some extent, represents a more recent model. Could this serve as evidence to demonstrate the effectiveness of our approach?

---

> > > ### Author Response · Authors · 2024-11-27
> > >
> > > Dear Reviewer, we have reviewed recent publications related to Knowledge Editing (KE) in an effort to demonstrate that GPT-J and LLaMA-2-7B are commonly used models in current research and represent the level of newer model architectures. You may refer to the following papers for further reference:
> > >
> > > WISE: Rethinking the Knowledge Memory for Lifelong Model Editing of Large Language Models （NeurIPS 2024）
> > >
> > > Retrieval-enhanced Knowledge Editing in Language Models for Multi-Hop Question Answering （CIKE 2024)
> > >
> > > PMET: Precise Model Editing in a Transformer (AAAI 2024)

---

> ### Author Response · Authors · 2024-11-23
>
> We are eager to know if you have any additional concerns or suggestions regarding our paper. If there are none, we sincerely hope you might consider raising the review score, and we would be more than willing to engage in detailed technical communication to address any potential concerns.

---

> ### Author Response · Authors · 2024-11-25
> **Request for Feedback Before Rebuttal Deadline**
>
> As we approach the rebuttal discussion deadline, we would like to follow up with you regarding the additional ablation experiments, generalization analysis, and writing updates we provided. Have these addressed your concerns? If so, we kindly ask if you would consider revising your review score accordingly.
>
> Thank you for your time and thoughtful evaluation of our work.

---

### Official Review · Reviewer_Pi7C · 2024-11-04

**Soundness:** 3
**Presentation:** 2
**Contribution:** 3
**Rating:** 6
**Confidence:** 3

**Summary:**

The paper addresses challenges in existing locate-then-edit knowledge editing methods, particularly focusing on multi-hop knowledge editing. The authors first investigate why locate-then-edit approaches struggle with multi-hop fact editing using tools from mechanistic interpretability, such as the logit lens and causal intervention. Their findings suggest that later MLP layers play a crucial role in storing multi-hop knowledge. To better update these knowledge-storing MLPs, the authors introduce a supplementary set construction method for each edit, transforming each edit into a multi-hop chain. This supplementary set then generates virtual multi-hop prompts specifically targeting the knowledge being edited. The authors utilize two-hop templates from this supplementary set to modify the model’s later layers, thereby enhancing multi-hop knowledge editing.

**Strengths:**

- **Comprehensive Analysis**: The paper presents a thorough analysis in Section 3, identifying that information related to multi-hop knowledge is stored in later MLP layers. Using interpretability tools such as the logit lens and causal interventions, the authors confirm this and propose edits to the last layer based on these findings.
- **Empirical Validation**: Experiments in Tables 3 and 4 show that the proposed method outperforms previous methods on multi-hop knowledge editing tasks, providing compelling evidence for its effectiveness.

**Weaknesses:**

- **Insufficient Presentation**: Section 5.2 lacks clarity as Tables 5 and 6 are referenced but not included in the main paper, instead being located in the appendix. While space constraints are understandable, omitting key results from the main discussion detracts from readability and coherence. Additionally, the absence of descriptions for the "base" in Tables 5 and 6 and missing details on the datasets used in Table 4 (e.g., whether MQuAKE is consistently used) leaves gaps in understanding.
- **Questionable Analysis of Results**: In Table 3, performance seems to decline post-editing compared to the original, but there is no discussion on the possible causes for this drop. If editing knowledge reduces performance, the rationale for the edits becomes unclear. It appears that “original” performance refers to accuracy on the unedited answer, but if so, why wasn’t accuracy on the correct answer also reported for the original model? Clearer explanations are necessary to clarify these performance metrics.
- **Limited Novelty in Method and Analysis**: The primary contribution of this work lies in the insight, obtained through interpretability tools, that multi-hop knowledge editing requires updating the later MLP layers. However, there is no clear experimental analysis showing that editing the later layers specifically contributes to performance improvement. It raises the question of whether merely expanding the support set was sufficient for better performance, implying that gains may not be attributed solely to the proposed method. To substantiate this claim, it would be useful to report the impact of the editing layers on performance, comparing different layers with and without the supplementary set.

**Questions:**

- **Figures**: Consider widening Figures 2(a) and 2(b) to improve readability and enlarge the subfigures in Figure 6 for better visibility.
- **Tables**: Provide a clearer explanation of the term “base” in Tables 5 and 6. It’s unclear whether this refers to the “original” in Table 3 or another baseline. Clear labels and descriptions would enhance understanding.

---

> ### Author Response · Authors · 2024-11-22
>
> Thank you for your positive feedback on our paper and for your valuable suggestions. Please find our detailed responses to the weaknesses and questions below.
> **Questions**
> > Q1: Figures: Consider widening Figures 2(a) and 2(b) to improve readability and enlarge the subfigures in Figure 6 for better visibility.
>
> The Figure 6 in previous version changes to Figure 7 in new new version. We have enlarged the subfigures in Figure 7(Figure 6 in previous version) to improve visibility. Due to space constraints, Figures 2(a) and 2(b) were not widened in the main text. However, larger versions are provided in the appendix, and a note has been added below the figures to inform readers.
>
> > Q2: Tables: Provide a clearer explanation of the term “base” in Tables 5 and 6. It’s unclear whether this refers to the “original” in Table 3 or another baseline. Clear labels and descriptions would enhance understanding.
>
> We use the term **Base** refers to the original model without any edits. And We revised the terminology in Tables and Figures, to consistently refer to the unmodified model as "Base". Additionally, we included a unified explanation of this term in the **Baseline** section of **5.1 Experimental Setup**.
>
> **Weakness**
> > W1: Insufficient Presentation: Section 5.2 lacks clarity as Tables 5 and 6 are referenced but not included in the main paper, instead being located in the appendix. While space constraints are understandable, omitting key results from the main discussion detracts from readability and coherence. Additionally, the absence of descriptions for the "base" in Tables 5 and 6 and missing details on the datasets used in Table 4 (e.g., whether MQuAKE is consistently used) leaves gaps in understanding.
>
> We use the term **Base** refers to the original model without any edits. And We revised the terminology in Tables and Figures, to consistently refer to the unmodified model as "Base" which is explained in the **Baseline** section of **5.1 Experimental Setup**. And add the dataset information in each table. Regarding the organization of the paper, we fully understand the difficulties you encountered when important results were placed in the appendix. We spent considerable time attempting to incorporate more tables into the main text but were ultimately unsuccessful. At the same time, we believe that the analytical section preceding the experiments is necessary. Therefore, we kindly ask for your understanding that due to space constraints, we had to place some tables, and even the entire ablation study, in the appendix. Our aim was to preserve the completeness of the experimental results as much as possible. All changes are highlighted in blue in the revised PDF file. We hope these revisions address your concerns comprehensively.

---

> ### Author Response · Authors · 2024-11-22
>
> > W2: Questionable Analysis of Results: In Table 3, performance seems to decline post-editing compared to the original, but there is no discussion on the possible causes for this drop. If editing knowledge reduces performance, the rationale for the edits becomes unclear. It appears that “original” performance refers to accuracy on the unedited answer, but if so, why wasn’t accuracy on the correct answer also reported for the original model? Clearer explanations are necessary to clarify these performance metrics.
>
>
> Yor are right. We report the **Base**'s performence on unedited answer. We believe this represents the upper limit of the model's inherent ability to utilize knowledge for answering multi-hop questions. We provided detailed explanations of the reported outcomes in tables in the **Setup and Hyperparameters** section of 5.1 **Experimental Setup**, ensuring consistent definitions for **"unedited answer"** and **"edited answer"**. Notes have also been added below each table to clarify the meaning of these values and report **Base**'s performence on edited answer too. We hope this explaination address your concerns about Questionable Analysis of Results in Table 3 and others.
>
> > W3: Limited Novelty in Method and Analysis: The primary contribution of this work lies in the insight, obtained through interpretability tools, that multi-hop knowledge editing requires updating the later MLP layers. However, there is no clear experimental analysis showing that editing the later layers specifically contributes to performance improvement. It raises the question of whether merely expanding the support set was sufficient for better performance, implying that gains may not be attributed solely to the proposed method. To substantiate this claim, it would be useful to report the impact of the editing layers on performance, comparing different layers with and without the supplementary set.
>
> To prove that editing the deeper layers specifically contributes to performance improvement. We have added a comprehensive ablation study and generalizability analysis in the Appendix section **E ABLATION STUDY AND GENERALIZABILITY OF IFMET**. These results demonstrate the positive contributions of the supplementary set and the interpretability-guided targeting of later MLP layers and the last token position. Importantly, they confirm that the observed performance improvements are not solely attributable to the supplementary set. You can refer to this section in new PDF for more detailed ablation experiment results.
>
> All changes are highlighted in blue in the revised PDF file. We hope these revisions address your concerns comprehensively.

---

> ### Author Response · Authors · 2024-11-23
>
> We are eager to know if you have any additional concerns or suggestions regarding our paper. If there are none, we sincerely hope you might consider raising the review score, and we would be more than willing to engage in detailed technical communication to address any potential concerns.

---

> ### Author Response · Authors · 2024-11-25
>
> As we approach the rebuttal discussion deadline, we would like to follow up with you regarding the additional ablation experiments, generalization analysis, and writing updates we provided. Have these addressed your concerns? If so, we kindly ask if you would consider revising your review score accordingly.
>
> Thank you for your time and thoughtful evaluation of our work.

---

> > ### Comment · Reviewer_Pi7C · 2024-11-25
> >
> > Thank you for your thoughtful response. After reviewing your answers to W1 and W2, my concerns regarding these points have been resolved. However, even after reading Appendix Section E, I find the analysis and results insufficient to fully address my question.
> >
> > In particular, could you provide a more detailed explanation of how the experimental results clearly demonstrate that the performance improvement is due to editing the later MLP layers, rather than merely increasing the support set? I understand that my delayed question may have left you with limited time to respond, and I sincerely appreciate your efforts despite the tight timeline.

---

> > > ### Author Response · Authors · 2024-11-25
> > > **Detailed explanation of ablation experiments**
> > >
> > > Firstly, we sincerely appreciate the time you've dedicated to reviewing our paper during this busy period. Below, we try to address your queries.
> > >
> > > Our interpretability analysis has identified that the existing editing methods fail to adequately modify knowledge in the deeper MLP layers, resulting in poor performance on multi-hop factual recall tasks. Additionally, our findings suggest that implicit multi-hop step dependencies rely on the knowledge provided by these deeper MLP layers. Based on these interpretability results at the last token position, we propose the IFMET. In the second stage of editing, we use a combination of supplement sets and modifications to the deeper MLP layers to update the knowledge therein.
> > >
> > > For specifics on the ablation study, please refer to the updated PDF attached.
> > > | Table index | Description |
> > > | -------- | -------- |
> > > | Table 7     | The results of the ablation experiments of MQuAKE-3K on GPT-J-6B model with edit batch=3000  |
> > > | Table 8 |  The results of the ablation experiments on GPT-J-6B model using a subset of MQuAKE-CF with edit batch=1 and 100   |
> > > | Table 9 |  The results of the ablation experiments on LLaMA-2-7B model using a subset of MQuAKE-CF with edit batch=1 and 100   |
> > >
> > > The three tables encompass various models and different edit batches, which we believe provide sufficient evidence to substantiate our claims. In all three tables, we have utilized the PMET as a baseline to assess method performance. The importance of each component is reflected through comparisons of performance improvements over PMET. PMET’s performance exemplifies a single-stage edit approach using shallow MLP edits based on single-hop edit query. In response to your  question, we particularly focus on two sub-experiments in our ablation study: "without supplementary set" (**w/o sup**) and "without deeper MLP modifications" (**w/o deeper**), as well as the final implementation of IFMET.
> > >
> > > * **IFMET**: In the second stage editing, we employ a multi-hop supplementary set alongside deep MLP editing techniques. Across all the experimental tables mentioned, IFMET consistently demonstrates a substantial improvement in inferential performance compared to PMET。
> > > * **w/o sup**: This represents that, in the second editing stage, we continued to use single-hop edit query instead of the supplement set to edit the deeper MLP layers. However, the results corroborate the interpretability analysis which emphasizes the differences between single-hop and multi-hop reasoning mechanisms. Compared to the original one-stage method PMET, performance fluctuations remained within a relatively stable range  in contrast to IFMET’s own +70% improvement. Therefore, we conclude that using single-hop data combined with deep MLP editing is ineffective, highlighting the critical importance of the supplementary set.
> > > * **w/o deeper**: However, as you suggest, is using only the supplement set sufficient? We can examine this through a specific sub-experiment. In this setup, the second-stage editing was modified to use the supplementary set combined with shallow MLP editing(rather than deeper MLP layers). If this also shows a significant performance improvement, it would indicate that merely expanding with the supplementary set, without considering its mechanism on the deeper MLP layers, can enhance results. However, as observed across the three tables, there was a consistent minor fluctuation in performance (ranging from -6.8% to +5.3%). In contrast to IFMET’s own +70% improvement, this underscores the importance of editing the deeper MLP layers when using the supplementary set.
> > >
> > > In light of the results from the ablation experiments **w/o sup** and **w/o deeper**, which align with our interpretability analysis, we emphasize that merely increasing the supplementary set is insufficient. It is essential to apply the supplementary set to the deeper MLP layers for knowledge editing to effectively enhance performance on multi-hop factual recall tasks.
> > >
> > > Please feel free to ask further questions at any time; there’s no need to hesitate. Additionally, if this detailed explanation resolves your queries, we will incorporate it into the updated version of the PDF.

---

> > > > ### Comment · Reviewer_Pi7C · 2024-11-26
> > > >
> > > > Thank you for the detailed response. Most of my concerns have been addressed during the rebuttal, and I have adjusted my score accordingly. Additionally, I request the authors to include their analysis in the main manuscript and resize the figures for better visibility in the camera-ready version.

---

> > > > > ### Author Response · Authors · 2024-11-26
> > > > >
> > > > > Thank you very much for recognizing our work and for the time you invested during the review period. We have incorporated more detailed analysis results into the paper and expanded Figures 2, 3, 4, 5, and 6 within the constraints of the page limit. These changes are included in the newly uploaded PDF. We will attempt further adjustments in the camera-ready version and greatly appreciate your feedback.

---

### Author Response · Authors · 2024-11-22
**General response**

**1.Index update**
The figure and table numbers have changed and some are newly added：


| Old index | New index
| -------- | -------- |
| Figure 6    | 7     |
| Table 5    | 12     |
| Table 6    | 11     |
| Table 7    | 17     |
| Table 8    | 13     |
| Table 9    | 14     |
| Table 10    | 15     |




| Old index | New index | Description |
| -------- | -------- | -------- |
| None    | Figure 6     | Causal Intervention result of MLP hidden state in last token position on LLaMA-2
| None    | Table 8     | Larger version of figure 2
| None    | Table 16     | the template of the prompt we used for asking LLaMA-2-7B to generate the supplementary set.

For improved visual clarity, we have widened the figure 2 and 7 as requested and corrected the typographical errors.


**2.Performance Correction**
For the editing performance of IFMET on the MQuAKE-3K dataset with an edit batch size of 3000 (as shown in Table 3), we did not perform multiple experiments to calculate the average due to time constraints during the initial submission. This has now been rectified through additional experiments, updating the value from 16.07 to 17.4.

**3.Polishing the Presentation**
In **5 Experiments** section, we use the term **Base** refers to the original model without any edits. And We revised the terminology in Tables and Figures, to consistently refer to the unmodified model as "Base". Additionally, we included a unified explanation of this term in the **Baseline** section of **5.1 Experimental Setup** and provide detailed explanations of the reported outcomes in tables in the **Setup and Hyperparameters** section of **5.1 Experimental Setup**, ensuring consistent definitions for "unedited answer" and "edited answer". Notes have also been added below each table to clarify the meaning of these values and report Base's performence on edited answer too.
All changes are highlighted in blue in the revised PDF file. We hope these revisions address your concerns comprehensively.

**4.Add Ablation Study and Generalizability Discussion Section**
More importantly, We have added a comprehensive ablation study and generalizability analysis in the Appendix section **E ABLATION STUDY AND GENERALIZABILITY OF IFMET**. These results demonstrate the positive contributions of the supplementary set and the interpretability-guided targeting of later MLP layers and the last token position. Importantly, they confirm that the observed performance improvements are not solely attributable to the supplementary set. Additionally, our experiments demonstrate that the proposed method can be extended to newer models such as LLaMA2. Furthermore, we have specifically explored the scalability of supplementary set construction, showing that it can be built by retrieving knowledge from within the model itself, eliminating the need for external knowledge bases.

---

### Meta-Review · Area_Chair_d6PD · 2024-12-19

**Metareview:**

This paper examines multi-hop knowledge editing in large language models, highlighting the shortcomings of current locate-then-edit methods. The authors' mechanistic interpretability analysis reveals that multi-hop knowledge queries process information differently from single-hop queries, with a greater reliance on deeper MLP layers. Current knowledge editing methods struggle with multi-hop queries as they primarily modify shallow layers while leaving deeper layers untouched. To address this, the authors introduce IFMET, a novel locate-then-edit approach that uses supplementary sets to convert single-hop edits into multi-hop chains, demonstrating enhanced performance on multi-hop knowledge editing tasks compared to existing methods.

While the paper offers valuable insights into how different layers handle single versus multi-hop knowledge, it falls short in several critical areas. The lack of clear ablation studies makes it impossible to determine whether performance improvements stem from deep layer editing or simply from expanded support sets. The paper fails to experimentally validate that editing deeper layers contributes to the observed gains, and its analysis of efficiency and complexity tradeoffs is inadequate.

The paper exhibits significant methodological and presentation flaws that warrant rejection. The central thesis regarding the importance of editing deeper layers for multi-hop knowledge lacks proper experimental validation, as improvements could be attributed to the expanded support set rather than the proposed layer-specific editing mechanism. The paper's presentation is problematic, with crucial results hidden in appendices, poorly explained performance metrics, and inconsistent result reporting. Furthermore, practical concerns arise from the method's dependence on external knowledge sources, limited testing on older models, and insufficient efficiency analysis.

For future consideration, the authors would need to conduct comprehensive ablation studies to validate deep-layer editing's specific contribution, enhance result presentation and clarity, expand testing to contemporary models, address implementation practicality, include detailed efficiency analysis, and strengthen the validation of key mechanistic claims. These requirements necessitate substantial additional research and validation rather than simple revisions.

**Additional Comments On Reviewer Discussion:**

The reviewers identified several key issues in the paper, and the authors made various attempts to address them. Regarding presentation, there were concerns about inconsistent numbering and unclear terminology. While the authors improved figure indexing and standardized terms around the "Base" model, these changes were primarily cosmetic and didn't address deeper methodological issues.

On the performance front, reviewers questioned result robustness. The authors acknowledged their initial limited experiments and conducted additional testing. The methodological validation was another significant concern, particularly the lack of ablation studies demonstrating component-specific contributions. Although the authors added new analysis in Appendix E, placing crucial validation information in an appendix rather than the main text remained problematic. Furthermore, the analysis failed to clearly distinguish between gains from supplementary sets versus layer-specific editing.

Questions arose about generalizability to newer models and reliance on external knowledge. The authors responded by adding LLaMA2 experiments and discussing model-based knowledge retrieval. However, they did not adequately address computational overhead analysis and practical implementation challenges.

These responses, while demonstrating an effort to address reviewer concerns, support a rejection recommendation for several reasons. The core methodological concerns about validating deep layer editing's specific contribution remain inadequately addressed, despite the addition of ablation studies. While presentation improvements were made, they were largely superficial, with critical analyses remaining in appendices instead of the main narrative. The generalizability additions showed some promise with LLaMA2 but failed to fully address practical implementation and efficiency concerns. Additionally, the performance corrections showing modest gains raised questions about the significance of the improvement given the added complexity.

Although the authors made sincere efforts to address reviewer feedback, the responses indicate that the paper's fundamental issues require substantial additional work beyond what could be addressed during the rebuttal period. The paper would benefit from a major revision focusing on stronger methodological validation and practical applicability before being suitable for publication.

---

### Decision · Program_Chairs · 2025-01-22

Reject